# Hematopoietic stem and progenitor cell-restricted Cdx2 expression induces transformation to myelodysplasia and acute leukemia

Therese Vu [1,2,12], Jasmin Straube [1], Amy H. Porter[1], Megan Bywater[1,2], Axia Song[1], Victoria Ling [1,2,3], Leanne Cooper[1], Gabor Pali[1], Claudia Bruedigam[1], Sebastien Jacquelin [1], Joanne Green[1], Graham Magor[4], Andrew Perkins [4], Alistair M. Chalk [5,6], Carl R. Walkley [5,6,7], Florian H. Heidel[8], Pamela Mukhopadhyay[1], Nicole Cloonan[1,13], Stefan Gröschel[9,10], Jan-Philipp Mallm [9], Stefan Fröhling [9,11], Claudia Scholl[9,11] & Steven W. Lane [1,2,3✉]

The caudal-related homeobox transcription factor CDX2 is expressed in leukemic cells but not during normal blood formation. Retroviral overexpression of Cdx2 induces AML in mice, however the developmental stage at which CDX2 exerts its effect is unknown. We developed a conditionally inducible Cdx2 mouse model to determine the effects of in vivo, inducible Cdx2 expression in hematopoietic stem and progenitor cells (HSPCs). Cdx2-transgenic mice develop myelodysplastic syndrome with progression to acute leukemia associated with acquisition of additional driver mutations. Cdx2-expressing HSPCs demonstrate enrichment of hematopoietic-specific enhancers associated with pro-differentiation transcription factors. Furthermore, treatment of Cdx2 AML with azacitidine decreases leukemic burden. Extended scheduling of low-dose azacitidine shows greater efficacy in comparison to intermittent higher-dose azacitidine, linked to more specific epigenetic modulation. Conditional Cdx2 expression in HSPCs is an inducible model of de novo leukemic transformation and can be used to optimize treatment in high-risk AML.

[1] Cancer Program, QIMR Berghofer Medical Research Institute, Brisbane, Australia. [2] School of Medicine, University of Queensland, Brisbane, Australia. [3] Cancer Care Services, Royal Brisbane and Women's Hospital, Brisbane, Australia. [4] Australian Centre for Blood Diseases, Monash University, Melbourne, Australia. [5] St. Vincent's Institute, Melbourne, Australia. [6] Department of Medicine, St. Vincent's Hospital, University of Melbourne, Fitzroy, Australia. [7] Mary MacKillop Institute for Health Research, Australian Catholic University, Melbourne, Australia. [8] Leibniz-Institute on Aging—Fritz Lipmann Institute (FLI) and Innere Medizin 2, Hämatologie und Onkologie, Universitätsklinikum Jena, Jena, Germany. [9] German Cancer Research Center, Heidelberg, Germany. [10] Department of Internal Medicine V, Heidelberg University Hospital, Heidelberg, Germany. [11] National Center for Tumor Diseases Heidelberg, Heidelberg, Germany. [12] Present address: University of Colorado Anschutz Medical Campus, Aurora, CO 80045, USA. [13] Present address: University of Auckland, Auckland, New Zealand. ✉email: Steven.lane@qimrberghofer.edu.au

The caudal-related homeobox gene *CDX2* is not expressed in normal hematopoietic stem cells (HSCs), but is expressed in ~90% of acute myeloid leukemia (AML) patients[1,2], as well as those with high-risk myelodysplastic syndrome (MDS) and advanced chronic myeloid leukemia (CML). Retroviral *Cdx2* expression in bone marrow (BM) progenitor cells facilitates in vitro self-renewal and causes a serially transplantable AML in vivo[1–3]. *CDX2* is thought to be necessary for leukemia growth, as knockdown of human *CDX2* by lentiviral-mediated short hairpin RNA (shRNA) impairs growth of AML cell lines and reduces clonogenicity in vitro[1]. These data indicate that aberrant *Cdx2* expression may promote HSC transformation to leukemia stem cells (LSCs).

*Cdx2* plays a critical role in embryogenesis and early developmental hematopoiesis[4–6]. Loss of *Cdx2* in murine blastocysts results in lethality at 3.5 days post-coitum[7]. *Cdx2* is a critical regulator of the trophectoderm layer, the first cell lineage to differentiate in mammalian embryos[8]. *Cdx2* downregulation in embryonic stem cells (ESCs) causes ectopic expression of the pluripotency markers *Oct4* and *Nanog*, while *Cdx2* upregulation triggers trophectoderm differentiation. *Cdx2* is also essential for in vitro trophoblast stem cell self-renewal, demonstrating a pivotal role for *Cdx2* in ESC fate specification[7–10]. In developmental hematopoiesis, *CDX2* and other caudal-related family members (*CDX1* and *CDX4*) are transcriptional regulators of homeobox (*HOX*) genes[11–13]. *HOX* gene function has been closely linked to self-renewal pathways in ESCs and HSCs, and the reactivation of these pathways by aberrant *HOX* expression has been implicated in leukemogenesis[14–17]. Despite this association, evidence of direct interaction between *CDX2* and the *HOX* cluster is lacking[18,19]. *CDX2* may also act via non-HOX pathways including via downregulation of *KLF4*[20,21]. Therefore, understanding targets of *CDX2* in hematological malignancy and mechanisms of transformation may provide new opportunities to treat patients with leukemia.

Retroviral overexpression models of oncogenesis provide a powerful tool to study the functional consequences of genetic mutations. However, these models also have limitations including the ex vivo manipulation of cells and preferential transduction of proliferative progenitor cells, rather than long-term HSCs. To overcome these barriers and to understand the mechanism of in vivo transformation of HSCs, we generated a transgenic model of *Cdx2* overexpression in hematopoietic stem and progenitor cells (HSPCs) to depict the cellular dynamics of transcriptional deregulation. Ectopic *Cdx2* expression in HSPCs results in lethal MDS, characterized by abnormal blood cell counts, dysgranulopoiesis, and thrombocytopenia, followed by secondary transformation to acute leukemia (AL) in a percentage of surviving mice. This is dependent on Cdx2 expression within HSPCs, as myeloid-restricted Cdx2 expression attenuates the phenotype. Unexpectedly, we observe reduced expression of *Hox* cluster genes and upregulation of differentiation factors in *Cdx2* HSPCs, signifying that non-Hox-mediated pathways drive these hematological diseases. *Cdx2*-driven leukemia is sensitive to azacitidine, with enhanced sensitivity when administered at a lower-dose on an extended schedule in comparison to a higher-dose on a shorter schedule. This work provides a model of MDS with stepwise transformation to AML that can be used to provide clinically relevant information for patients with MDS and AML with multilineage dysplasia.

## Results

### Ectopic expression of *Cdx2* alters function of HSPCs.
To examine the effects of *Cdx2* expression in adult hematopoiesis, we generated a transgenic mouse by insertion of *Cdx2* (NCBI gene ID: 12591) and *mCherry* (Clontech) open reading frames downstream from a CAG promoter and a loxP-flanked stop cassette in the mouse *Rosa26* locus of C57BL/6 ES cells (LSL-*Cdx2*-mCherry, TaconicArtemis). The *Cdx2* and *mCherry* cDNAs were separated by a T2A self-cleaving peptide, which allowed for co-expression of the two proteins after Cre excision of the stop cassette between the loxP sites. Thus, *mCherry* reported expression of *Cdx2* in cells following Cre-recombinase mediated activation (Supplementary Fig. 1a). The LSL-*Cdx2*-mCherry mice were crossed to *Scl*-CreER[T] mice[22] to generate offspring that can inducibly express *Cdx2-mCherry* in HSCs following tamoxifen exposure (Scl:Cdx2; Supplementary Fig. 1b). Scl:Cdx2 and control mice (Ctrl; consisting of mice from the genotypes: C57BL/6 wild-type [WT], Scl-CreER[T], and LSL-Cdx2-mCherry) were fed a diet of rodent chow containing tamoxifen (400 mg/kg) for two weeks. *Cdx2* expression was confirmed in mCherry-positive BM cells by western blot on whole BM and by quantitative reverse transcriptase PCR (qRT-PCR) (Supplementary Fig. 1c, d). Scl:Cdx2 mice showed mCherry expression by flow cytometry at two weeks, and this rose further by four weeks after tamoxifen (Fig. 1a). To evaluate Cdx2 expression differences between previously published retroviral models[1] and Scl:Cdx2 transgenic cells, we transduced *Scl*-CreER[T] lineage-negative BM with MSCV-IRES-GFP (MIG)-Cdx2 and MIG-Empty retrovirus. Retroviral CDX2 overexpression resulted in approximately 900-fold higher CDX2 expression than Scl:Cdx2 transgenic cells, potentially accounting for phenotypic differences (Supplementary Fig. 1d).

Scl:Cdx2 mice showed decreases in B220-positive B cells and CD3-positive T cells, and a concomitant increase in Gr1-positive myeloid cells in PB compared with control mice (Fig. 1b) and independent of total PB leukocyte count at week four (Supplementary Fig. 1e). In BM, common lymphoid progenitors (CLP; lineage$^{low}$IL7Rα$^+$) were unaffected by Cdx2 expression in HSC (Supplementary Fig. 1f), as were double-negative T cell populations (DN1-4, Supplementary Fig. 1g). However, a significant loss of B cell progenitors was observed (Supplementary Fig. 1h), indicating a B lymphocyte differentiation block in Scl:Cdx2 BM. Furthermore, *mCherry*-positive cells were prominent in HSC-enriched LKS + (lineage$^{low}$c-Kit$^+$Sca-1$^+$) and granulocyte-macrophage progenitors (GMP; lineage$^{low}$c-Kit$^+$Sca-1$^-$CD16/32$^+$CD34$^+$), but lower in megakaryocyte-erythroid progenitors (MEP; lineage$^{low}$c-Kit$^+$Sca-1$^-$CD16/32$^-$CD34$^-$) and more mature lineage$^{low}$ cells (Fig. 1c). The frequency of GMPs was increased in Scl:Cdx2 mice compared with controls, while common myeloid progenitors (CMP; lineage$^{low}$c-Kit$^+$Sca-1$^-$CD16/32$^{mid}$CD34$^+$) and MEPs were significantly decreased (Fig. 1d–g). Decreased absolute CMP and MEP cell counts were also observed in Scl:Cdx2 BM, despite Scl:Cdx2 BM being normocellular (Supplementary Fig. 1i–l). These data validate the model, demonstrating that *Cdx2* is preferentially expressed in immature HSPC subsets, causing skewing towards the granulocyte lineage.

*Cdx2* expression in HSPC also led to depletion of long-term hematopoietic stem cells (LTHSCs [LKS$^+$CD150$^+$CD48$^-$]) and short-term HSCs (STHSCs [LKS$^+$CD150$^-$CD48$^-$]) in Scl:Cdx2 BM (Fig. 1h, i, Supplementary Fig. 1m). In vitro colony forming cell (CFC) assays with BM cells four weeks after tamoxifen induction demonstrated enhanced colony formation after two weeks (Fig. 1j), with enrichment of *mCherry*-positive cells at each passage (Supplementary Fig. 1n) but *Cdx2* expression did not facilitate in vitro serial replating beyond two weeks. These LTHSCs exclusively harbor self-renewing potential[23], implying that the cellular mechanism of LTHSC exhaustion might involve enforced cell cycle entry and loss of quiescence. To test this hypothesis, we performed in vivo

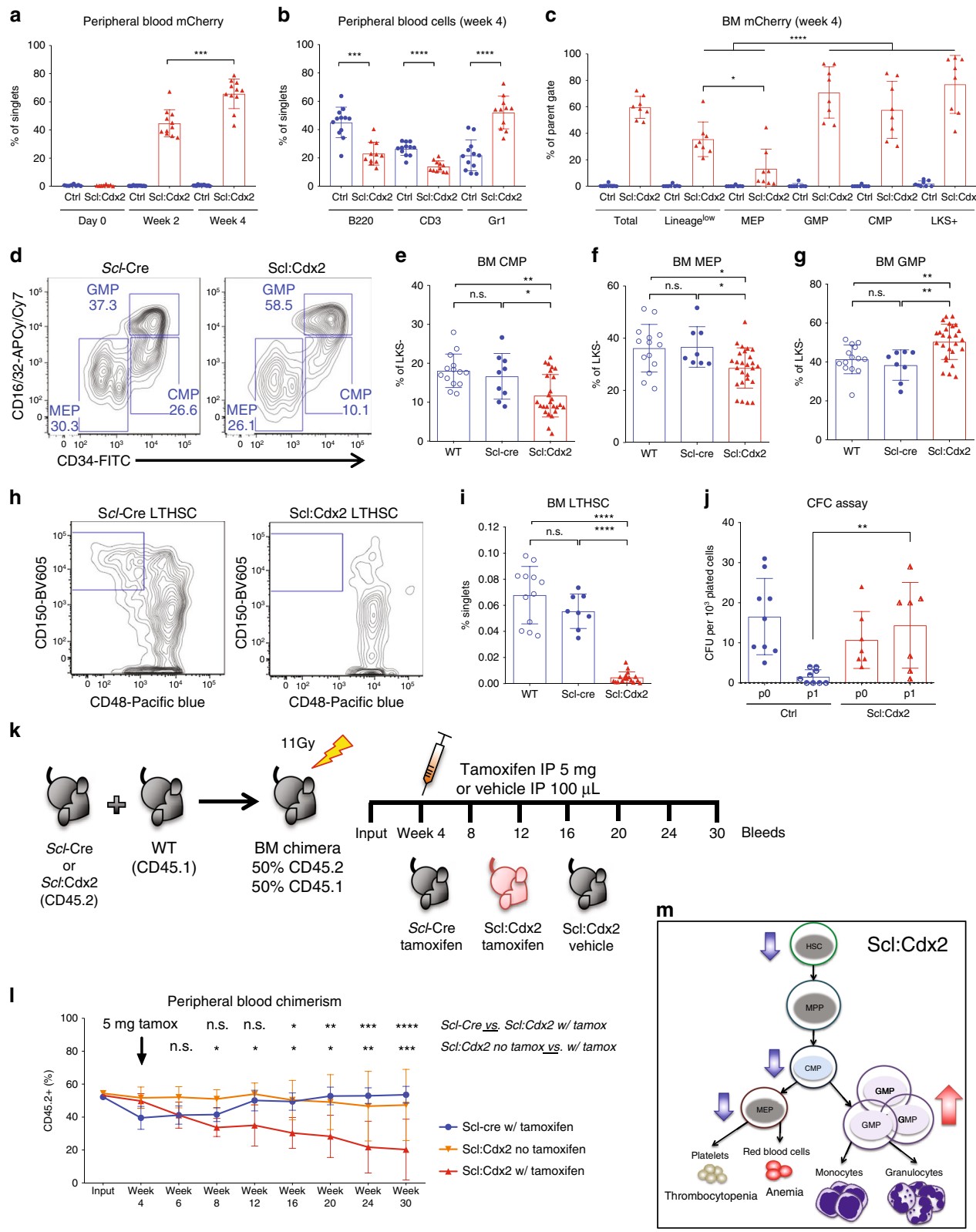

competitive BM transplantations using Scl-CreER[T] or Scl:Cdx2 donor BM (expressing CD45.2) from uninduced (naïve) mice, mixed with congenically marked competitor wild type (WT) BM (expressing CD45.1) (Fig. 1k). There was equivalent engraftment of donor cells of both genotypes four weeks after transplantation in the absence of tamoxifen. Induction of *Cdx2*

expression following intraperitoneal (IP) injection of 5 mg of tamoxifen caused a progressive loss of PB chimerism (Fig. 1l), which was associated with reduced BM HSPC populations (LKS+ and LTHSC) in induced mice transplanted with Scl:Cdx2 BM (Supplementary Fig. 1o–r). Altogether, these data indicate that cell-intrinsic expression of *Cdx2* impairs HSC

**Fig. 1 Cdx2 expression in HSPC alters progenitor subsets and self-renewal function. a** Frequency of mCherry-positive Cdx2 cells in peripheral blood (PB) after tamoxifen induction at indicated time points (Ctrl $n = 12$; Scl:Cdx2 $n = 11$). **b** Frequency of mCherry-positive Cdx2 cells in B220 + , CD3+ or Gr1 + PB cells at week 4 after tamoxifen (Ctrl $n = 12$; Scl:Cdx2 $n = 11$). **c** Frequency of mCherry-positive Cdx2 cells in bone marrow (BM) at week 4 in indicated subpopulations ($n = 8$ per group). **d** Representative flow cytometry of lineage$^{low}$cKit$^+$Sca1$^-$ (LKS−) myeloid progenitors in BM cells at week 4 after tamoxifen. (**e**) Frequency of common myeloid progenitors (CMP), **f** megakaryocyte-erythroid progenitors (MEP), and **g** granulocyte-macrophage progenitors (GMP) in parent population (LKS−), (WT $n = 14$; Scl-cre $n = 8$; Scl:Cdx2 $n = 26$). **h** Representative flow cytometry of lineage$^{low}$cKit$^+$, Sca1$^+$ (LKS+) BM cells showing CD150$^+$CD48$^-$ long-term hematopoietic stem cells (LTHSC) and (**i**) quantification of LTHSC frequency, (WT $n = 12$; Scl-cre $n = 8$; Scl:Cdx2 $n = 19$). **j** Colony forming cell (CFC) assay of BM cells initially plated (p0) and replated (p1) in M3434 methylcellulose. Each BM sample was plated in triplicate and each data point represents the mean of triplicate plates (Ctrl $n = 9$; Scl:Cdx2 $n = 7$). **k** Diagram of BM transplant experiment setup. Scl-Cre ($n = 5$) and Scl:Cdx2 ($n = 10$, split into $n = 5$ per treatment arm) BM chimeras. Tamoxifen or corn oil (vehicle) was administered by intraperitoneal (IP) injection to indicated groups. **l** PB chimerism to monitor relative contribution of Scl-Cre or Scl:Cdx2 BM to peripheral hematopoiesis. Experiment was performed in duplicate. Arrow indicates IP injection time point. (**m**) Model of Scl:Cdx2 hematopoietic cell hierarchy showing decreases in LTHSC, CMP and MEP leading to a loss of platelets (thrombocytopenia) and erythrocytes (anemia), and a relative increase in GMP resulting in greater levels of myeloid cells: monocytes and granulocytes. $N =$ biologically independent animals. Statistical analyses performed using two-tailed Mann–Whitney test except (**l**) which used mixed-effects model with Tukey's multiple comparisons test. Data are plotted as mean values $+/-$ SD. n.s.; not significant. $*P < 0.05$, $**P < 0.01$, $***P < 0.001$, $****P < 0.0001$.

function with reduced capacity to sustain long-term hematopoiesis (Fig. 1m).

**Cdx2 expression in HSPCs induces MDS and AL**. After tamoxifen induction of Cre-recombinase, Scl:Cdx2 mice developed a variety of hematological diseases including MDS, myeloproliferative neoplasm (MPN) and AL (Fig. 2a, b). Scl:Cdx2 mice had a median survival of 43 weeks, while no disease was seen in Scl-CreER$^T$ controls (Fig 2a, b, Supplementary Fig. 2a). MDS was evidenced by reduced blood counts together with reticulocytosis, fragmented erythrocytes, anisopoikilocytosis, and neutrophil dysplasia (Fig. 2a). MPN was characterized by leukocytosis, reticulocytosis, and hypersegmented neutrophils (Fig. 2a, c, d). AL was diagnosed by >20% blasts in PB and BM (Fig. 2a, Supplementary Fig. 2b), together with leukocytosis, splenomegaly, and anemia (Fig. 2c, d, g). All moribund mice had reduced hemoglobin compared with controls (Fig. 2e) while all Scl:Cdx2 mice (regardless of health state) showed mild to profound thrombocytopenia (Fig. 2f, Supplementary Fig. 2c). All Scl:Cdx2 mice showed a propensity for hypersegmented neutrophils, and expansion of Gr1-positive myeloid cells and decrease in B220 B cells compared with Scl-CreER$^T$ controls (Supplementary Fig. 2d). Approximately 20% of Scl:Cdx2 mice did not develop overt hematological disease (Fig. 2b) aside from thrombocytopenia and neutrophil dysplasia.

In mice that developed AL, we observed biphasic disease, with initial MDS (dysplasia, leukopenia and thrombocytopenia; Supplementary Fig. 2h, i) followed by the later onset of leukocytosis, anemia, and increased mCherry+ and c-Kit+ cells in PB (Supplementary Fig. 2j–m). Immunophenotyping revealed distinct leukemia lineage commitments (Fig. 2h). Scl:Cdx2 #252 showed a clonal expansion of c-Kit$^+$B220$^{int}$CD3$^{int}$ cells (Supplementary Fig. 2e), Scl:Cdx2 #882 PB leukemic cells were c-Kit$^+$CD3$^+$mCherry$^+$, representative of acute T-cell leukemia (Supplementary Fig. 2f), but most mice (#2259, #2261, and #472) developed acute myeloid/erythro-myeloid leukemia with a c-Kit$^+$mCherry$^+$ population predominately Gr1$^+$CD11b$^-$ (Supplementary Fig. 2g). The evolution of MDS to AL in Scl:Cdx2 mice (Supplementary Fig. 2h-j) with an expansion of mCherry-expressing c-Kit+ cells (Supplementary Fig. 2k-m) is likely due to the acquisition of transformation events and is consistent with secondary leukemia after MDS observed in patients. The leukemias were transplantable as irradiated recipient mice phenocopied the primary donor in all cases (example in Fig. 2i) and had shortened survival compared with the primary setting (Fig. 2j), demonstrating rapid expansion of the leukemic clone.

Taken together, these data show that Cdx2 is able to transform HSPC populations in situ into a faithful model of MDS with secondary AML.

**Secondary genetic lesions cooperate with Cdx2 expression**. AML transformation is mediated through co-operative mutations in genes that confer a proliferative advantage to cells together with pathways that primarily impair cellular differentiation[24]. To determine whether co-operating mutations had contributed to Scl:Cdx2 HSPC full transformation, we performed whole exome sequencing (WES) of three AL samples and one MPN sample. WES was performed on genomic DNA of CD45.2-sorted cells (ie. donor cells) from transplanted leukemic mice. Tumor samples were sourced from mCherry-positive donor cells and compared with germline samples that were mCherry-negative donor cells. We found a number of frameshift and non-synonymous somatic mutations in known tumor-associated genes, including positive (Jak1, Raf1, Zap70) and negative regulators (Pten, Cgref1) of signal transduction, cell adhesion molecules (Fat1), transcription factors (Etv6, Ikzf1, Trp53), and DNA-binding proteins (Nabp2) (Supplementary Table 1). PTEN is a known tumor suppressor commonly altered in human AML[25], and was mutated in Scl:Cdx2 AML along with ETV6, a recurring fusion partner with CDX2[2,26]. Other AML single nucleotide variants (SNVs) were uncovered in Fat1 and Raf1. Loss-of-function of cadherin-like protein Fat1 and mutations in the Ras effector Raf1 are also previously described in AML[27,28]. Bilineage ALL cells harbored a frameshift insertion in Ikzf1 zinc-finger protein, which is frequently mutated in human B-ALL and to a lesser extent in T-ALL[29–32]. Mutations in the tyrosine kinase Jak1 (sample #252), are more prevalent in T-ALL than B-ALL and are associated with poor prognosis[33,34]. Finally, Cdx2-induced erythro-myeloid leukemia #472 harbored a loss of heterozygosity (LOH) event in the commonly mutated tumor suppressor gene Trp53. To further determine the significance of these SNVs, we confirmed their presence in functional protein domains similar to pathogenic SNVs in human orthologues[35] (examples shown in Supplementary Fig. 3a, Supplementary Data 1).

We did not observe any SNVs in cancer-associated genes (as listed in MSK-HemePACT cancer panel and COSMIC[36]) from Scl:Cdx2 MPN BM demonstrating that the emergence of secondary mutations was found exclusively in AL.

Using the Beat AML trial cohort[37], we found significant co-expression of CDX2 and FLT3 in AML patients, as well as increased CDX2 expression in FLT3-internal tandem duplication (ITD)-positive samples compared with FLT3-ITD-negative

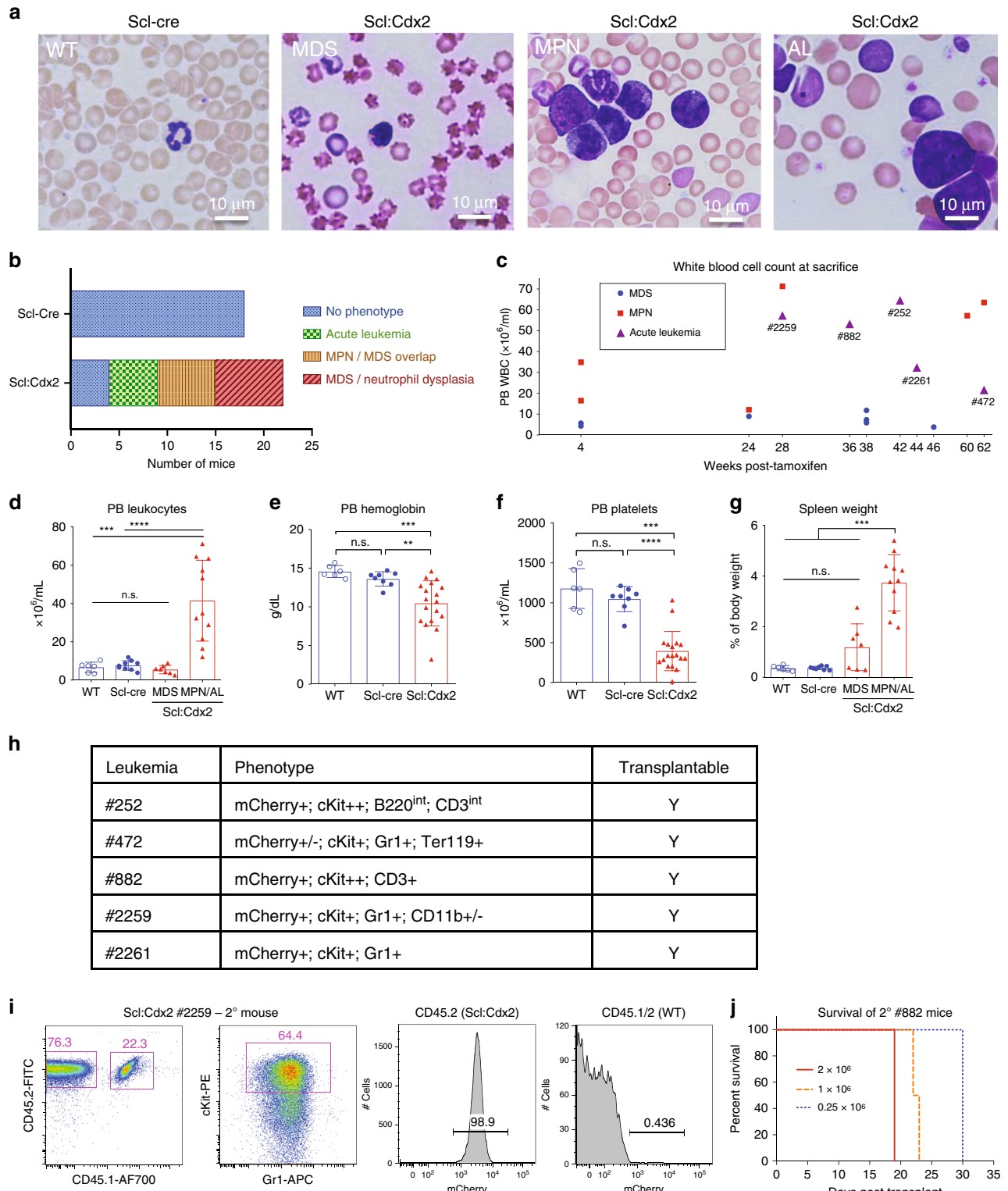

**Fig. 2 Cdx2 expression in HSPC results in lethal and transplantable disease. a** Representative Wright-Giemsa stained PB smears showing distinct phenotypes of *Scl*-Cre (normal) and Scl:Cdx2 mice at disease onset. Scale bars on bottom right of each image indicate 10 μm. MDS, myelodysplastic syndrome; MPN, myeloproliferative neoplasm; AL, acute leukemia. **b** Incidence of hematological disease in *Scl*-Cre (*n* = 18) and Scl:Cdx2 (*n* = 22) mice after 60 weeks of monitoring. (**c**) White blood cell (WBC) counts of Scl:Cdx2 mice (*n* = 18) at date of sacrifice and assignment of disease diagnosis. The identification numbers of mice with acute leukemia are indicated. **d** Leukocyte counts (WBC), (**e**) hemoglobin, **f** platelet counts, and **g** spleen size relative to body mass of mice at sacrifice. (WT *n* = 6; *Scl*-cre *n* = 8; Scl:Cdx = 18 [MDS *n* = 7; MPN/AL *n* = 11]). **h** Table showing acute leukemia immunophenotype in mCherry+cKit+ PB cells of Scl:Cdx2 mice at disease onset and transplantability status of these cells. **i** Representative flow cytometry of PB from mice transplanted with Scl:Cdx2 #2259 acute leukemia BM cells. **j** Survival curve of secondary leukemia mice transplanted with varying cell numbers of Scl:Cdx2 #882 BM cells (*n* = 10). N = biologically independent animals. Statistical analyses performed using two-tailed Mann–Whitney test. Log-rank Mantel-Cox test used for survival curve. Data are plotted as mean values +/− SD. n.s.; not significant. *P < 0.05, **P < 0.01, ***P < 0.001, ****P < 0.0001.

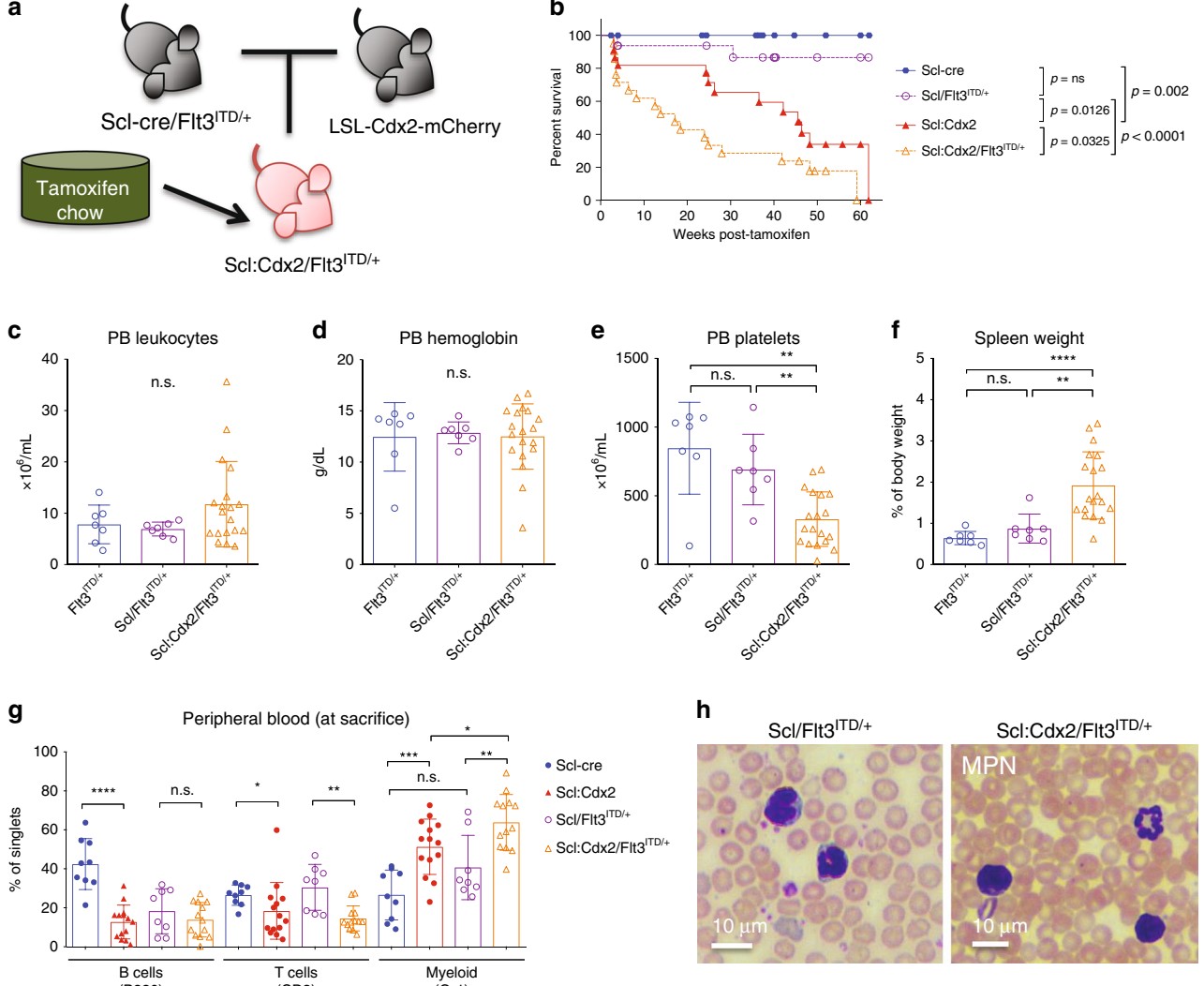

**Fig. 3 Cdx2 synergizes with Flt3-ITD to accelerate myeloproliferation. a** Diagram of breeding strategy to produce Scl:Cdx2/Flt3$^{ITD/+}$ mice. LSL-Cdx2-mCherry mice were crossed with Flt3$^{ITD/+}$ germline mutant mice that were previously crossed with *Scl*-CreER$^T$ transgene mice (Scl-cre/Flt3$^{ITD/+}$). Unexcised Scl:Cdx2/Flt3$^{ITD/+}$ mice are fed tamoxifen-loaded chow for two weeks. **b** Survival curve of tamoxifen-treated mice (*Scl*-CreER$^T$ $n = 18$; Scl:Cdx2 $n = 22$; Scl/Flt3$^{ITD/+}$ $n = 16$; Scl:Cdx2/Flt3$^{ITD/+}$ $n = 21$). **c** PB leukocytes (WBC), **d** hemoglobin and **e** platelet counts of mice, and (**f**) spleen size relative to body mass of mice at sacrifice (Flt3$^{ITD/+}$ $n = 7$; Scl/Flt3$^{ITD/+}$ $n = 7$; Scl:Cdx2/Flt3$^{ITD/+}$ $n = 19$). **g** Frequency of B220-positive B cells, CD3-positive T cells and Gr1-positive myeloid cells in PB of mice at sacrifice (Scl-cre $n = 9$; Scl:Cdx2 $n = 14$; Scl/Flt3$^{ITD/+}$ $n = 8$; Scl:Cdx2/Flt3$^{ITD/+}$ $n = 13$). **h** Representative Wright-Giemsa stained smear of Scl/Flt3$^{ITD/+}$ PB with mild granulopoiesis and Scl:Cdx2/Flt3$^{ITD/+}$ PB showing monocytosis and hypersegmented neutrophils resulting in MPN (observed in multiple independent animals). Scale bars on bottom left of each image indicate 10 μm. $N$ = biologically independent animals. Statistical analyses performed using two-tailed Mann–Whitney test. Log-rank Mantel-Cox test used for survival curve. Data are plotted as mean values +/− SD. n.s.; not significant. *$P < 0.05$, **$P < 0.01$, ***$P < 0.001$, ****$P < 0.0001$.

samples (Supplementary Fig. 3b, c). We therefore tested whether Scl:Cdx2 mice would accelerate development of AML when crossed with mice harboring Flt3-ITD, a common oncogene in AML[38] (Fig. 3a). Scl:Cdx2/Flt3$^{ITD/+}$ double mutant mice had shorter survival and disease latency compared with Scl:Cdx2 mice and Scl/Flt3$^{ITD/+}$ alone (Fig. 3b). There was a trend to leukocytosis in some Scl:Cdx2/Flt3$^{ITD/+}$ mice compared with controls (Fig. 3c, Supplementary Fig. 3d). Hemoglobin levels showed a wide range across biological replicates, however there was no significant difference between the means of Scl:Cdx2/Flt3$^{ITD/+}$ and controls (Fig. 3d). Scl:Cdx2/Flt3$^{ITD/+}$ double mutants showed severe thrombocytopenia (Fig. 3e) and succumbed to advanced MPN (Fig. 3b) characterized by splenomegaly (Fig. 3f) and increased in Gr1$^−$positive myeloid cells in PB compared with control or single knockin Cdx2 mice (Fig. 3g).

Significant myeloid skewing in Scl:Cdx2/Flt3$^{ITD/+}$ mice was only observed after tamoxifen induction (Supplementary Fig. 3e, f). While Scl:Cdx2/Flt3$^{ITD/+}$ mice had monocytosis, increased granulopoiesis, and hypersegmented neutrophils (Fig. 3h) they did not develop AML. Overall, these data demonstrate that ectopic *Cdx2* expression collaborates with *Flt3*$^{ITD/+}$ in hematopoietic cells to accelerate lethal myeloid disease but is insufficient for AML transformation.

**Cdx2 limited to myeloid progenitors attenuates disease.** *Cdx2* expression in HSPCs is leukemogenic and results in the expansion of GMPs (Figs. 1g, 2a). We used *LysM*-Cre (*Lyz2*-Cre) to express *Cdx2* in GMP and terminally differentiated myeloid cells, but not HSCs to test whether Cdx2 expression in only myeloid

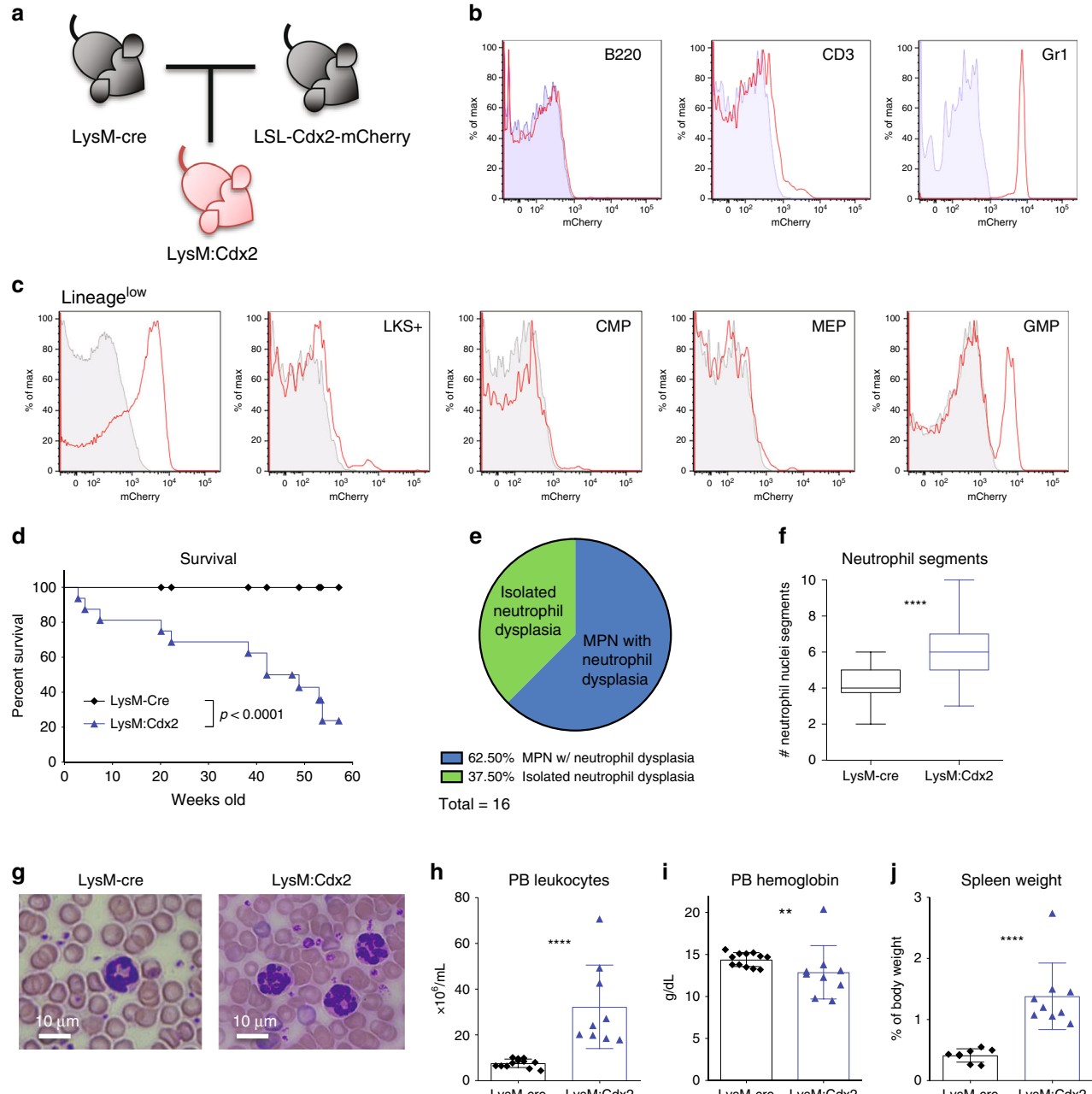

**Fig. 4 Restricted myeloid expression of Cdx2 causes disease distinct from Scl:Cdx2. a** Diagram of breeding strategy to product LysM:Cdx2 mice. LSL-Cdx2-mCherry mice are crossed with *LysM*-Cre knock-in mice at the endogenous *LysM* locus, leading to LysM-expressing GMP cells and their progeny to express Cdx2 and mCherry-reporter proteins. **b** Representative flow cytometry plots showing mCherry expression in PB subsets of *LysM*-Cre (blue shaded) and LysM:Cdx2 (red line) mice. **c** Representative flow cytometry plots showing mCherry expression in BM subsets of *LysM*-Cre (grey shaded) and LysM:Cdx2 (red line) mice. **d** Survival curve after 60 weeks of monitoring (*LysM*-Cre, $n = 23$; LysM:Cdx2, $n = 16$). **e** Proportion of hematological state of LysM:Cdx2 mice at sacrifice. **f** Box plot of quantification of neutrophil nuclei segments in PB smears. Top and bottom of whiskers represent maximum and minimum values respectively, centre line within box represents median value, upper and lower quartiles represent 75th and 25th percentile respectively. At least 15 neutrophils were counted per slide by independent researchers, $n = 4$ per group. **g** Wright-Giemsa stained PB smears showing representative neutrophils of *LysM*-Cre and LysM:Cdx2 mice. Scale bars on bottom left of each image indicate 10 μm. **h** PB leukocyte (WBC) and **i** hemoglobin levels of mice at sacrifice (*LysM*-Cre $n = 12$; LysM:Cdx2 $n = 9$). **j** Spleen size relative to body mass of mice at sacrifice (*LysM*-Cre $n = 7$; LysM:Cdx2 $n = 9$). $N =$ biologically independent animals. Statistical analyses performed using two-tailed Mann–Whitney test. Data are plotted as mean values $+/-$ SD. n.s.; not significant. **$P < 0.01$, ****$P < 0.0001$.

cells is sufficient for AML (Fig. 4a). *mCherry* expression was exclusive to myeloid cells from the GMP stage onwards, and not in CMPs, HSCs or MEPs that are not myeloid-committed (Fig. 4c). Moreover, mCherry was only present in terminally differentiated Gr1-positive cells in PB and absent in B220-positive and CD3-positive lymphoid cells (Fig. 4b). LysM:Cdx2 mice

succumbed to a long-latency MPN (Fig. 4d, e). LysM:Cdx2 mice had significant neutrophil hypersegmentation compared with LysM-Cre controls (Fig. 4f, g), with leukocytosis, anemia, and splenomegaly (Fig. 4h–j), but unlike Scl:Cdx2 mice, did not exhibit thrombocytopenia (Supplementary Fig. 4a) and did not develop secondary AML. White blood cell (WBC) counts of

LysM:Cdx2 mice showed progressive leukocytosis and myeloid skewing with age (Supplementary Fig. 4b–d). Together, these data demonstrate that the cell of origin of *Cdx2* expression fundamentally affects the subsequent hematological phenotype.

**Cdx2 drives differentiation and impairs self-renewal**. To gain insight into the mechanism underlying Cdx2-mediated transformation of HSPCs, we performed RNA sequencing (RNA-Seq) on Scl-Cre, Scl:Cdx2, Scl/Flt3[ITD/+], and Scl:Cdx2/Flt3[ITD/+] mice (outlined in Supplementary Table 2, Supplementary Data 2). Four weeks after tamoxifen, LKS + cells were flow purified (mCherry-positive in Scl:Cdx2 mice, Supplementary Fig. 5a). As expected, the top upregulated gene was *Cdx2* (Fig. 5a). Principal component analysis (PCA) revealed clustering based on genotype (Fig. 5b). Gene set enrichment analysis (GSEA) revealed the loss of self-renewal markers[39] in *Cdx2*-expressing cells (Fig. 5c), suggesting that Scl:Cdx2 leads to a loss of HSC function, increased proliferation and progenitor cell differentiation. This was supported by the enrichment of committed progenitor cell signatures within mCherry-positive Scl:Cdx2 HSPCs compared with Scl-Cre (Fig. 5d, e, Supplementary Tables 3 and 4). Consistent with previous reports showing that LSC rely on oxidative phosphorylation for their survival[40], there was enrichment of mitochondrial metabolism pathways and pathways associated with malignant progression (Fig. 5f–h, Supplementary Table 3) suggesting that Scl:Cdx2 cells share fundamental cellular processes with LSCs. Furthermore, cell cycle analysis in Scl:Cdx2 LKS + cells demonstrated fewer cells in quiescent $G_0$ phase and increased $G_1$ phase compared with control samples (Fig. 5i, Supplementary Fig. 5b). Interestingly, Scl:Cdx2 LKS+ cells were less apoptotic (Fig. 5j, Supplementary Fig. 5c). In summary, *Cdx2* drives a dominant gene expression program within HSPCs that leads to a proliferative, progenitor-cell like phenotype.

Unexpectedly, there was downregulation of *Hox* cluster genes in Scl:Cdx2 HSPCs (Supplementary Fig. 5d). Other groups have shown increased expression of *Hox* genes after enforced *Cdx2* overexpression[1,3], however the specific *Hox* gene targets were non-overlapping. In the context of normal HSC function, decreased *Hox* gene function is associated with loss of self-renewal[41,42] and progressive downregulation of *Hox* genes is seen in normal differentiation (Supplementary Fig. 5e)[43]. We therefore hypothesized that *Cdx2* may bind to factors that regulate myeloid differentiation, leading to concomitant downregulation of *Hox* genes in stem cell populations. To understand the regulatory activity of Cdx2 within rare HSPCs, we utilized Assay for Transposase Accessible Chromatin with high-throughput sequencing (ATAC-Seq) on purified Scl:Cdx2 HSPCs vs. controls (*Scl*-CreER[T] alone), to identify changes in chromatin accessibility mediated by Cdx2[44]. In total, 62,711 peaks were identified in Cdx2-expressing cells and 28,282 peaks in the *Scl*-CreER[T]. The majority of chromatin accessible regions (26,099) were shared between both groups. These common regions were dominated by promoter elements whereas condition-specific regions were dominated by distal elements (Fig. 6a, Supplementary Fig. 6b). Within the Scl:Cdx2 specific distal elements, we found the Cdx2 motif ($p = 1.6e^{-34}$) centrally enriched and also motifs belonging to the CCAAT/enhancer-binding protein family (Cebpb [$p = 9.8e^{-89}$], Cebpe [$p = 1.6e^{-89}$], Cebpa [$7.5e^{-83}$], Cebpd [$p = 9.2e^{-89}$]) (Fig. 6b) confirmed with another algorithm (HOMER,[45] Supplementary Fig. 6c). We also compared our data to the publicly available CEBPα ChIP-Seq dataset (GSM1187163) performed in GMP and found a significant overlap in peaks in Scl:Cdx2 BM but not *Scl*-Cre control samples (Fig. 6c, Supplementary Fig. 6d). ATAC-Seq provides representation of Cdx2 binding and suggests that *Cdx2* expression associates with chromatin changes that increase the accessibility of pro-differentiation myeloid

transcription factor binding sites of the CCAAT/enhancer-binding protein family.

Coordinated RNA-Seq and ATAC-Seq data provide evidence of transcriptional and epigenetic reprogramming of leukemic stem cell populations. ATAC-Seq showed enrichment of early myeloid progenitor programs in pre-leukemia samples, with progressive acquisition of committed megakaryocyte erythroid progenitor chromatin architecture in erythroid leukemia, and lymphoid chromatin architecture in lymphoid leukemias, even though these cells retained a stem cell surface immunophenotype (Supplementary Fig. 6e, f). Furthermore, we used RNA-Seq profiles of each Cdx2-expressing leukemia to identify differentially expressed genes that were upregulated in T-ALL (#882) and B/T-ALL (#252) but not other samples (Supplementary Data 3). Here using the tool Enrichr[46,47], we found significant enrichment for genes deregulated upon transcription factor alteration in T lymphocytes and T cell leukemia ($p < 0.05$), again showing lymphoid priming within the stem cell populations (Supplementary Data 4). These data are consistent with a *Cdx2*-induced transcriptional program priming LKS + towards progenitor cell differentiation. In support of this, RNA-Seq also showed upregulation of *Cebp* family genes in Scl:Cdx2 LKS+ (representing pre-leukemic HSPC, Supplementary Fig. 5f) in keeping with myeloid differentiation. Interestingly, transformed Scl:Cdx2 LKS+ BM cells from acute leukemic mice showed similar or decreased levels of *Cebp* gene transcripts compared with control cells, with the sole exception of *Cebpb* (Supplementary Fig. 5f), suggesting these leukemia cells downregulate effectors of myeloid commitment as a mechanism of transformation.

Next, we performed chromatin immunoprecipitation sequencing (ChIP-Seq) to identify Cdx2 binding sites in hematopoietic cells. We validated MSCV-IRES-GFP-Cdx2[1] tagged with a FLAG epitope (Cdx2-FLAG) by immunoprecipitation with rabbit anti-FLAG monoclonal antibody and confirmed expression and binding of FLAG-tagged Cdx2 in Ba/F3 cells (Supplementary Fig. 6a). We next transduced lineage-negative WT mouse BM with Cdx2-FLAG or empty vector (EV) and performed ChIP-Seq. Cdx2-FLAG ChIP-Seq confirmed strong central enrichment of Cdx2 motifs at peaks in both promoter and distal regions (Fig. 6b). To overcome any dilution of binding signal of Cdx2-expressing HSPC, we sought to integrate Cdx2-FLAG ChIP-Seq data from lineage-negative cells with ATAC-Seq on LKS+ and publicly available CEBPα ChIP-Seq on GMP. The top 1000 gained peaks in either Scl:Cdx2 or *Scl*-Cre ATAC-Seq showed correlation with Cdx2-FLAG and CEBPα ChIP-Seq peaks (Fig. 6c), suggesting these cell populations share similar chromatin identity despite immunophenotypic differences. To further functionally assess the relevance of the Scl:Cdx2 gained or lost distal accessible chromatin regions in HSPCs and myeloid progenitors, we analyzed ATAC-Seq and histone methylation marks associated with enhancers (H3K4me1) and ChIP-Seq data of LKS+ (or MPPs), CMPs, and GMPs[43]. Scl:Cdx2 HSPCs had less accessibility at enhancer regions (regions that are ATAC-accessible or histones with H3K4me1 modification) that regulate fate in LKS/MPP cells (Fig. 6c, d; Supplementary Fig. 6g, h). In contrast, Scl:Cdx2 HSPCs show an increase in accessibility of distal enhancer peaks that regulate committed myeloid progenitor cell differentiation (Fig. 6c, d; Supplementary Fig. 6g, h). These data suggest that *Cdx2* results in chromatin remodeling at distal enhancers, with a bias towards increased accessibility of enhancers associated with myeloid differentiation and reduced accessibility at enhancers of cell types with self-renewal potential. Concordantly, ChIP-Seq revealed Cdx2 peaks at *HoxA* and *HoxB* loci consistent with Cdx2 binding (Fig. 6e) and correlation with transcriptional downregulation of *Hox* family genes

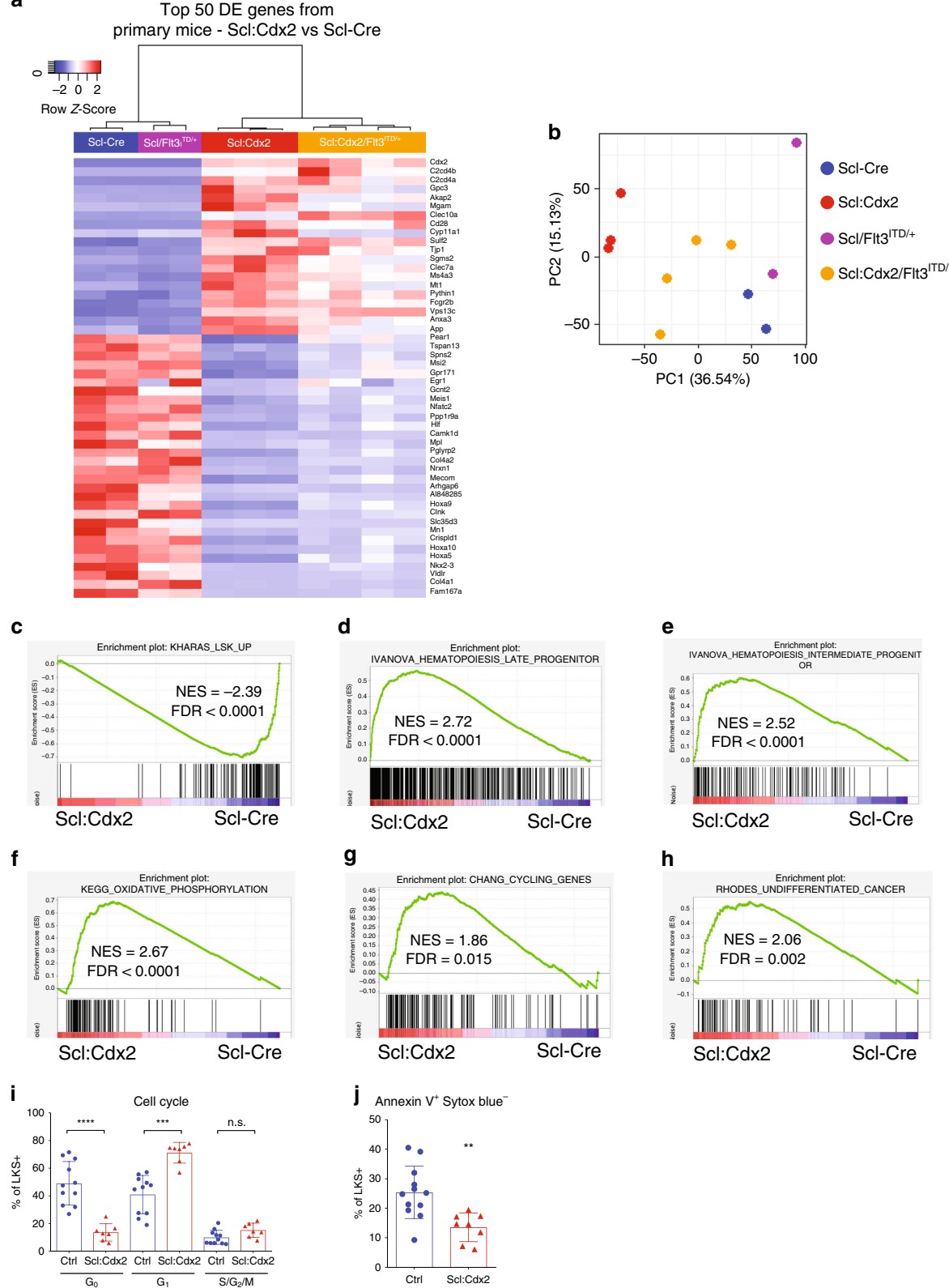

**Fig. 5 Cdx2 confers a progenitor gene signature associated with differentiation. a** Heatmap and unsupervised hierarchical clustering of top 50 differentially expressed genes from comparing Scl:Cdx2 and *Scl*-Cre on normalized read counts from RNA-Seq of BM LKS + cells of *Scl*-Cre, Scl:Cdx2, Scl/Flt3$^{ITD/+}$, and Scl:Cdx2/Flt3$^{ITD/+}$ mice. Color scale shows Z-score after row normalization. **b** Principal component analysis of the total RNA-Seq derived transcriptome. **c–h** Gene Set Enrichment Analysis (GSEA) on normalized read counts of transcripts from RNA-Seq of *Scl*-Cre ($n = 2$) compared with Scl:Cdx2 ($n = 3$). **i** Cell cycle analysis of BM LKS+ cells (*Scl*-cre $n = 11$; Scl:Cdx2 $n = 7$). **j** Apoptosis analysis of Annexin V-positive, Sytox-negative cells in BM LKS + by flow cytometry (*Scl*-cre $n = 12$; Scl:Cdx2 $n = 8$). N = biologically independent animals. Statistical analyses of graphs performed using two-tailed Mann–Whitney test. Data are plotted as mean values $+/-$ SD. n.s.; not significant. **$P < 0.01$, ***$P < 0.001$, ****$P < 0.0001$.

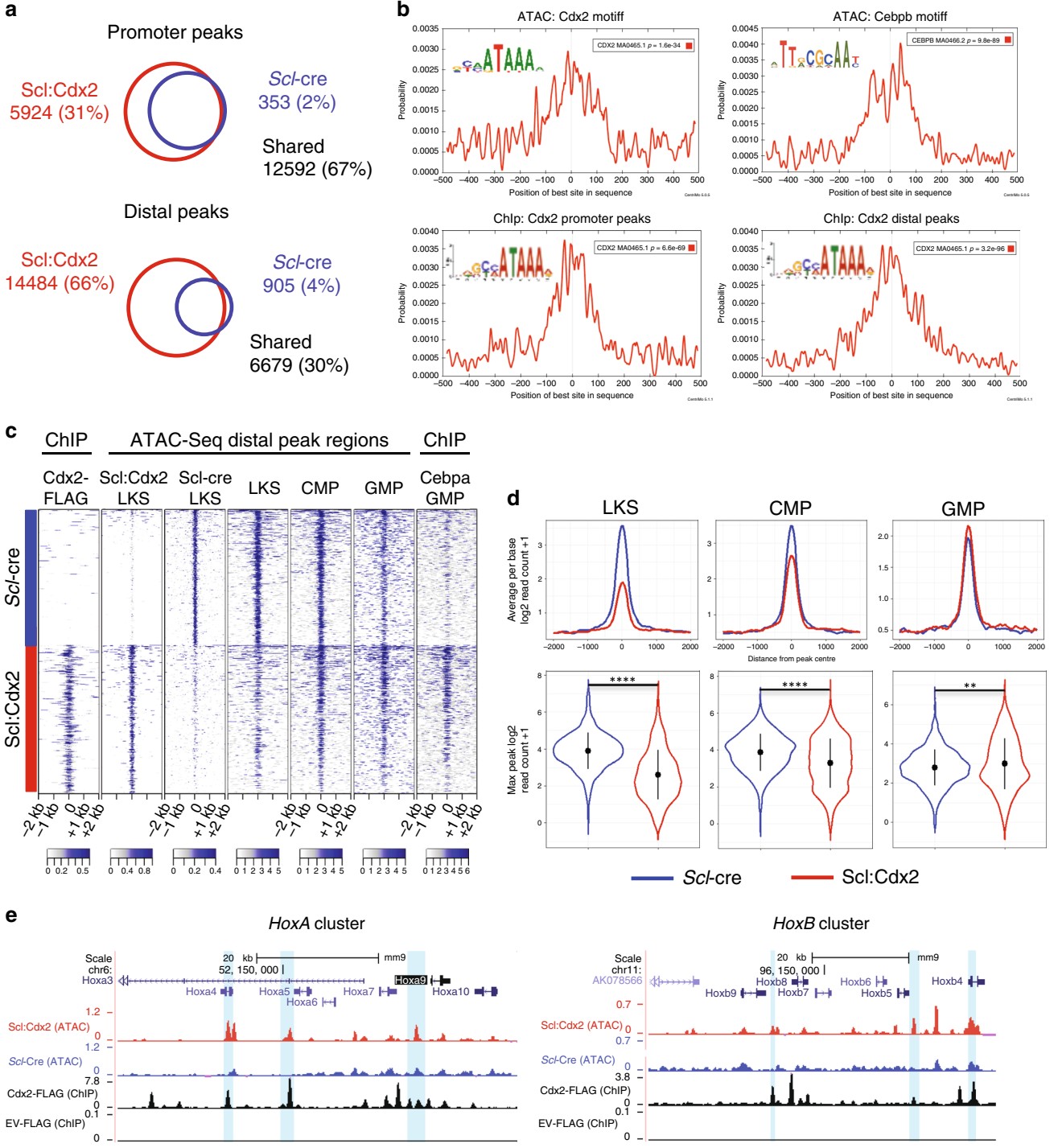

**Fig. 6 Cdx2 modifies chromatin access in regions with critical differentiation factors. a** Number of ATAC-Seq peaks found in *Scl*-Cre and Scl:Cdx2 BM LKS+ cells. **b** Centrally enriched motifs in peaks specific for Scl:Cdx2 BM samples by ATAC-Seq and Cdx2-FLAG BM samples by ChIP-Seq (as labeled) determined by MEME Suite. **c** Heatmaps of the top 1000 gained peaks at distal elements (peak centres +/− 2000 base pairs) that overlap between Cdx2-Flag BM samples by ChIP-Seq and Scl:Cdx2 or *Scl*-Cre BM samples by ATAC-Seq, ordered by signal intensity. LKS, CMP, GMP ATAC-Seq read coverage[43] and Cebpα GMP ChIP-Seq read coverage included at the same locations. Color scale shows Log2 + 1 normalized read counts. **d** Average peak height (top) and max peak height distribution (bottom) of distal ATAC-Seq peaks relating to (**c**). **e** UCSC browser tracks (mm9 murine genome) of Scl:Cdx2 and *Scl*-Cre ATAC-Seq peaks, and Cdx2-FLAG and Empty vector (EV) ChIP-Seq peaks in *HoxA* and *HoxB* clusters. Peaks called by MACS2. Light blue vertical bars highlight Scl:Cdx2 gained peaks positioned with a Cdx2 ChIP-Seq peak and Cdx2 motif. Statistical analyses for (**d**) performed using two-tailed Mann–Whitney test without adjustment for multiple comparisons. **P < 0.01, ****P < 0.0001.

(Supplementary Fig. 5d). Together these data suggest that *Cdx2* represses certain *Hox* genes and primes HSPCs for myeloid differentiation.

**Cdx2 leukemia is sensitive to myeloid disease therapy.** Scl: Cdx2-induced MDS with secondary transformation to AML (sAML) is mediated by common oncogenic mutations seen in human disease, and thus, this model provides an opportunity to examine the preclinical efficacy of anti-leukemic drugs. sAML is refractory to standard chemotherapy and is associated with dismal survival. We performed preclinical studies to evaluate the activity and in vivo mechanism of 5-azacitidine (Aza), a clinically approved therapy for high-risk MDS and AML[48]. We evaluated mice that had received secondary transplants from Scl:Cdx2 AML, together with support WT BM cells. Aza treatment commenced once donor engraftment was established at 2 mg/kg IP injection daily for one week followed by three weeks of rest (Supplementary Fig. 7a), mimicking the clinical schedule[49]. After one cycle of treatment, we observed a dramatic reduction in WBC counts in Aza-treated mice but not in vehicle treated controls (Fig. 7a). This was supported by similar pronounced reduction in mCherry cells and c-Kit expression in PB of Aza mice (Fig. 7b, c). In all experiments, the leukemia relapsed by the end of cycle one, however a second cycle of Aza dosing led to reduced leukemic burden and significant improvement in overall survival (Fig. 7d). There was increased apoptosis of Aza-treated Cdx2 cells, showing direct cytotoxicity of Aza on leukemia cells, with minimal effects on WT support cells or vehicle treated mice (Fig. 7e, Supplementary Fig. 7b). We also compared the standard 7 day regimen (2 mg/kg, 14 mg total per cycle) to a lower dose of Aza administered for 14 days over a 28 day cycle (1 mg/kg, qd, Monday-Friday, 14 mg total per cycle) (Supplementary Fig. 7c), to mimic the prolonged exposure to low level drug that is seen with oral Aza dosing[50,51]. Interestingly, greater improvement was seen in mice receiving low exposure, extended duration (LE-ED) with Aza, compared with high exposure, limited duration (HE-LD) Aza (Fig. 7f, g). These data were confirmed in an independent myeloid leukemia model, also driven by *Cdx2* (#2261) (Fig. 7h, i), suggesting that dose and scheduling may be relevant in optimizing clinical responses to Aza in MDS/AML.

RNA-Seq was performed on mCherry-positive LKS+ cells from mice treated with vehicle vs. HE-LD and LE-ED Aza (Supplementary Fig. 7d, Supplementary Data 5). LE-ED Aza treatment enriched for gene signatures associated with DNA hypomethylation (Fig. 7j), in accordance with mechanistic changes supported through extended oral dosing of Aza[50,51]. In contrast, HE-LD Aza treatment enriched for DNA damage and apoptosis signatures suggesting cytotoxicity of this regimen (Fig. 7k, Supplementary Fig. 7e, f). Both groups of Aza-treated cells showed significant upregulation of Trp53 and downregulation of Mycn (Fig. 7l), supporting a general mechanism of Aza in the induction of p53 and suppression of cellular proliferation[52,53]. The gene expression changes seen after Aza treatment mimicked the signature found in *Cdx2* expressing cells prior to AML transformation (Fig. 7m), suggesting that Aza may revert AML to a pre-leukemic state, and also upregulates *Klf4* (Fig. 7l), a gene known to be repressed by *Cdx2* and has been shown to have a tumor suppressor function in AML[21]. Altogether, these data demonstrate the preclinical efficacy of Aza in MDS/AML and suggest that extended schedules of low-dose therapy may have improved efficacy compared with standard regimens.

## Discussion

Transcriptional deregulation is a common leukemic mechanism that is thought to perturb cellular self-renewal and differentiation by modifying developmental cues. *CDX2* is essential for ESC fate determination and is aberrantly expressed in myeloid malignancy. We generated a conditional transgenic mouse model of *Cdx2* activation and characterized the de novo phenotype of *Cdx2* expression in various hematopoietic subsets. Mice expressing *Cdx2* in HSPCs develop lethal hematological diseases with prominent features of MDS and subsequent transformation into AL. The development of AL shows a long clinical latency with stepwise acquisition of oncogenic mutations, suggesting that Cdx2 expression predisposes cells to a pre-leukemic state with permissive conditions to the accumulation of cooperating secondary genetic events. This closely reflects the progression of human MDS to AML, where stepwise genetic mutations occur within HPSC and identifies these immature populations as the reservoir for leukemia initiating activity in vivo. Importantly, this model allows temporal control of *Cdx2* expression within HSCs, leading to in situ transformation of HSPCs to LSCs, thereby eliminating the confounding effects of ex vivo manipulation of HSPC populations and retroviral models.

In humans, ectopic *CDX2* expression is described in AML but also approximately 80% of newly diagnosed ALL or pediatric ALL[54,55], underscoring the clinical relevance of this model. Unexpectedly, our model shows strong downregulation of *Hox* factors in fully transformed leukemia, which contrasts with other studies[3], and suggests that *Cdx2* can activate a number of discrete oncogenic pathways for leukemogenesis. We suggest that CDX2 expression correlates differently with HOX expression in different contexts. For example, expression levels of CDX2 are comparable in ALL and AML samples[3], however HOX deregulation is much less common in ALL than AML. Furthermore, in embryogenesis, Cdx2 coordinates posterior development via Hox-independent mechanisms[56]. In keeping with other publications[21], we frequently observed repression of *Klf4* in all cases of AL.

When *Cdx2* is expressed in HSPCs, mice show a propensity to develop secondary mutations followed by the development of a range of ALs of varying lineages. Conversely, when *Cdx2* expression is restricted to myeloid cells in LysM:Cdx2 mice, there is a more homogeneous phenotype, typified by myelocytic expansion, leukocytosis, and splenomegaly, but without the thrombocytopenia that is hallmark to Scl:Cdx2 mice. Transformation to leukemia was not observed in this model, consistent with the hypothesis that HSPCs represent a leukemia-initiating seed population that is required for full disease penetrance.

As mCherry is not observed in LysM:Cdx2 MEPs, platelet numbers are not affected in these mice. In contrast, hypersegmented neutrophils are present in both LysM:Cdx2 and Scl:Cdx2 models, suggesting that *Cdx2* expression within GMP cells is key to this phenotype. In addition, *Cdx2* expression at the HSPC level is also seen to affect lymphoid lineages, highlighting the multipotent nature of Scl:Cdx2 cells. Emphasizing this, we observe a key regulatory role of aberrant *Cdx2* on common hematopoietic developmental pathways.

BM characterization of moribund pre-leukemic Scl:Cdx2 mice show a relative decrease in MEPs and relative increase in GMP compared with controls. This phenotype may represent differentiation arrest at this cellular level and is consistent with reports from human high-risk MDS patients[57] such as MDS with excess blasts. Initially, we observed a modest increase in in vitro colony formation of Scl:Cdx2 BM but no immortalization. In competitive BM transplant assays, Scl:Cdx2 cells had cell-autonomous HSC self-renewal defects. These data indicate that *Cdx2* leads to impaired clonogenicity, a trait that is similar to other animal models of MDS mutations[58].

Pathological changes in high-risk MDS cells may confer apoptotic resistance and provide growth and survival advantages, leading to leukemia progression[59]. This is consistent with the

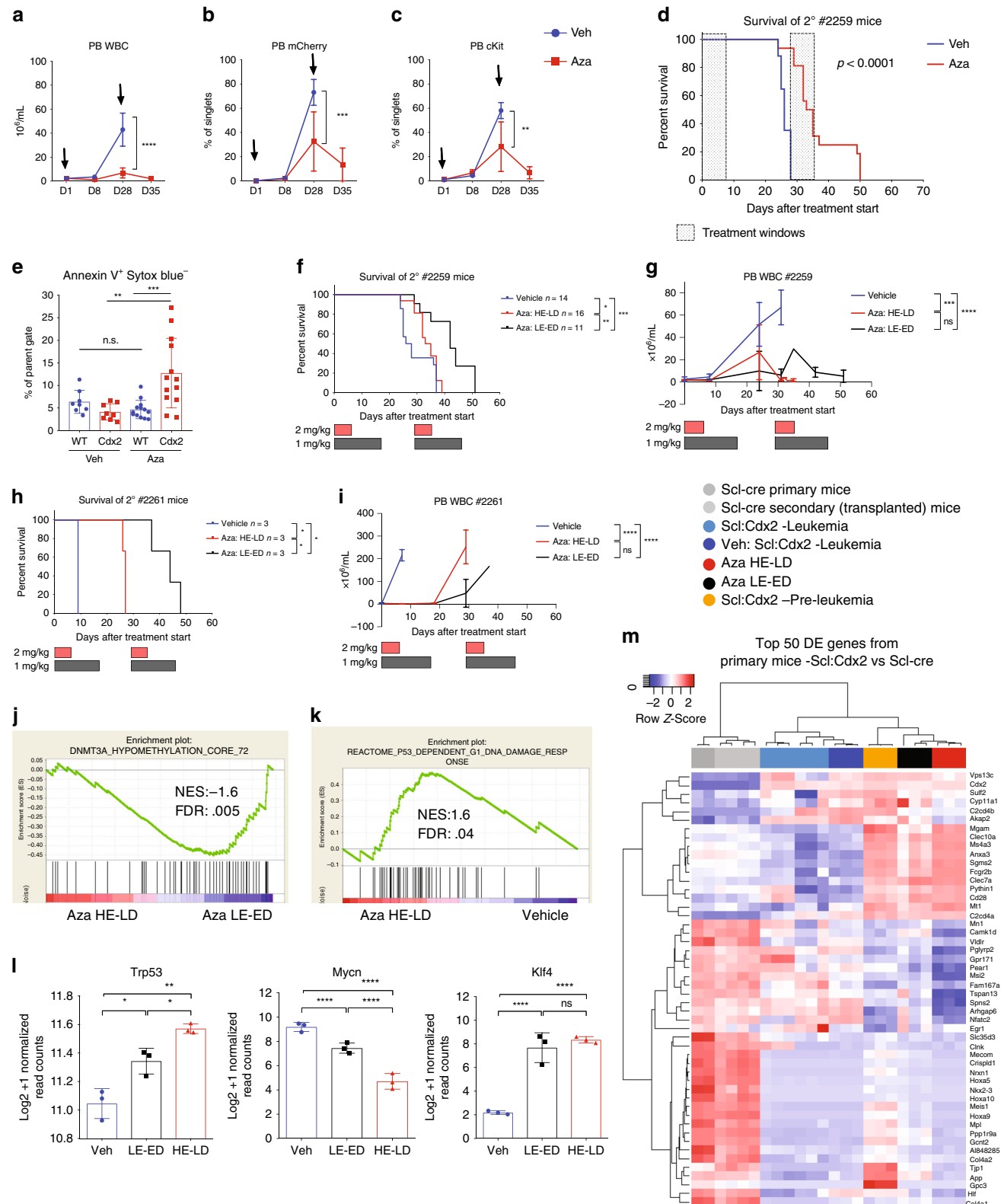

**Fig. 7 Cdx2-mediated leukemia is responsive to treatment with 5-azacitidine.** Time course of PB (**a**) WBC count, (**b**) mCherry percentage and (**c**) immature cKit cells of Scl:Cdx2 #2259 leukemia mice treated with Vehicle (Veh) n = 8 or Azacitidine (Aza) n = 9. Black arrows denote start of each 7 day treatment cycle. **d** Survival curve of Veh n = 17 or Aza n = 16. **e** Apoptosis assay of CD45.1+ WT or CD45.2+ Scl:Cdx2 PB cells treated with Veh or Aza. **f** Survival curve of Scl:Cdx2 #2259 mice treated with Veh (n = 14), Aza for 7 days at 2 mg/kg (HE-LD) per dose (n = 16) or Aza for 14 days at 1 mg/kg (LE-ED) per dose (n = 11). Red shaded bars denote HE-LD Aza treatment window. Black shaded bars denote LE-ED Aza treatment window. **g** PB WBC count time course. **h** Survival curve of Scl:Cdx2 #2261 mice treated with Veh, Aza HE-LD or Aza LE-ED (n = 3 per group). **i** PB WBC count time course. GSEA of (**j**) hypomethylation signature and (**k**) DNA damage signature (n = 3 per group). **l** Normalized read counts of transcripts by RNA-Seq (n = 3 per group). **m** Unsupervised hierarchical clustered heatmap derived from RNA-Seq of top 50 differentially expressed genes from comparing Scl:Cdx2 and Scl-Cre. Visualized are Scl-cre (gray) representing normal LKS+, Scl:Cdx2 leukemia cells (light blue) and cells treated with vehicle (dark blue) representing Cdx2-mediated acute leukemia, Scl:Cdx2 pre-leukemia cells (orange), and Scl:Cdx2 leukemia cells treated with HE-LD Aza (red) or LE-ED (black). Color scale shows row-normalized Z-score. N = biologically independent animals. Statistical analyses of graphs performed using multiple t-tests (one unpaired t-test per row) with few assumptions except (**l**) which used FDR-adjusted P-values from likelihood ratio test of the binomial generalized log-linear modeled gene expression data as implemented in edgeR. Log-rank Mantel-Cox test used for survival curves. Data are plotted as mean values +/− SD. n.s.; not significant. *P < 0.05, **P < 0.01, ***P < 0.001, ****P < 0.0001.

---

leukemic transformation[64–66]. Mice bearing the heterozygous activating mutation Flt3$^{ITD/+}$ develop a non-lethal MPN resembling chronic myelomonocytic leukemia (CMML) without transformation to AML[38]. When crossed with Scl:Cdx2 mice, we observe that double mutant Scl:Cdx2/Flt3$^{ITD/+}$ dramatically accelerate, lethal MDS/MPN. We speculate that the shorter disease latency of Scl:Cdx2/Flt3$^{ITD/+}$ mice potentially affects the chances of oncogenic progression seen in Scl:Cdx2 counterparts.

Advanced MDS and leukemic transformation has traditionally been challenging to model in animals. For this reason, studies into the use of azacitidine in MDS have largely come from primary patient samples[67]. In transplant experiments of Scl:Cdx2 secondary leukemias, we find that Aza prolongs survival of mice compared with vehicle treated controls, and Aza is preferentially toxic to Cdx2-mCherry-positive cells. Using dosing schedules comparable to CC-486 oral Aza regimens used in human clinical trials[50,51], Aza appears to be more effective and more specific for hypomethylating genes when administered in a lower-dose, extended schedule compared with higher-dose, limited schedule. These preclinical findings warrant follow-up clinical trials, for example, through the use of extended schedules of oral Aza in patients with MDS that do not respond to standard Aza. Our data suggest that Aza alone is insufficient to deplete LSC as all mice relapsed after 1-2 cycles of treatment. This is consistent with the clinical scenario and it is likely that combination strategies (for example, with venetoclax[68]) may be required to induce meaningful long-term remissions.

Altogether, this work characterizes a model of conditional Cdx2 expression that demonstrates transformation of normal HSPCs to MDS and AL in situ. Cdx2 alters HSPC identity and confers pre-leukemic progenitor cell characteristics, facilitating clonal evolution with important biological correlates of human leukemia. This model can be used to study the clinical effects of Aza, and demonstrates that prolonged, low doses of hypomethylating agents may increase specificity and efficacy of these agents against MDS and AML.

## Methods

**Animals and phenotypic analysis.** Experimental animals were maintained on a C57BL/6J strain in a pathogen-free animal facility and procedures were approved by the QIMR Berghofer Animal Ethics Committee (A11605M). Mice were housed in clean cages with shredded tissue as nesting material, and environmental enrichment provided as often as possible. Cages were maintained at an ambient temperature of 20–26 °C on a 12 h light/dark cycle. LSL-Cdx2-mCherry mice were generated by TaconicArtemis. Flt3$^{ITD/+}$ mice[38] were obtained from Dr. Wallace Langdon, Perth. Scl-CreER$^T$ mice[22] were obtained from Dr. Carl Walkley, Melbourne. LysM-Cre mice were obtained from Jackson Laboratories. Azacitidine was dissolved in 0.9% saline by vortexing for 60 seconds and injected intra-peritoneally within two hours. Any remaining solution was discarded after use due to short-term stability of the drug. Peripheral blood (PB) was collected by retro-orbital venous blood sampling into EDTA-coated tubes and analyzed on a Hemavet 950

analyzer (Drew Scientific). PB smears were prepared and stained with Wright-Giemsa (BioScientific) according to the manufacturer's protocol. Twenty microliters of fresh PB was lysed with 1 mL Pharmlyse (BD Biosciences) and stained with B220, CD33, Gr1, Mac1, and c-Kit for 15–30 min at 4 °C. Flow cytometric data collection was performed on a fluorescence-activated cell sorter LSRII Fortessa (BD Biosciences) with BD FACSDiva software (version 8.0.1) and analyzed using FlowJo (version 9.9.6). Flow cytometry antibodies were used at 1:100 dilution unless otherwise specified (Supplementary Table 5). BM cells were harvested by flushing femur and tibia bones. LKS + (Lineage$^{low}$cKit$^+$Sca1$^+$) cells were stained as previously described[69]. In brief, cells were stained with a lineage cocktail comprising of biotinylated antibodies (B220, CD3e, CD5, Gr1, Mac1, Ter-119). Cells were then stained with Streptavidin, c-Kit and Sca1. Common myeloid progenitors (CMP), granulocyte-macrophage progenitors (GMP) and megakaryocyte-erythroid progenitors (MEP) cells were identified with the addition of CD34 and CD16/32. Short-term (ST-) and long-term hematopoietic stem cells (LTHSC) were stained with the addition of CD48 and CD150. Incubations were performed for 20-30 min at 4 °C. For sorting, cells were purified using a FACSAriaIII (BD Biosciences). Cell cycle analysis was performed by staining cells with surface markers for LKS+ followed by fix and permeabilization according to the manufacturer's instructions (Fix & Perm kit, Thermo Fisher). Cells were stained with Ki-67 (B56) (1:100) in permeabilization buffer for 30 min at 4 °C. Cells were washed and resuspended in PBS with Hoechst 33342 (20 μg/mL, Invitrogen) prior to flow cytometry analysis. Events were acquired at <1000 events/s. Apoptosis analysis was performed by staining cells for LKS+ markers and keeping incubation times to 15 min to minimize cell death. Washed cells were then stained with 2.5 μL Annexin V (Biolegend) in 50 μL Annexin V binding buffer (BD Biosciences) (1:20) for 15 min in the dark at room temperature. Cells were not washed and 250 μL of Annexin V binding buffer containing 0.25 μL of Sytox blue (Invitrogen) was added. Cells were analyzed by flow cytometry within one hour.

**Competitive BM transplantation.** BM cells derived from 6- to 8-week-old Scl-CreER$^T$ or Scl:Cdx2 mice (1 × 10$^6$ cells, expressing CD45.2) were combined with equal numbers of age-matched Ptprca CD45.1 congenic competitor BM cells, and injected into the lateral tail vein of lethally irradiated (11 Gy total in two separate fractions at least 3 h apart) 6- to 8-week-old C57BL/J × Ptprca CD45.1/CD45.2 congenic recipient female mice (animals sourced from Animal Resource Centre, Western Australia).

**Colony forming assay.** BM cells were washed with PBS and seeded into 1 mL of methylcellulose (M3434; Stem Cell Technologies) in 35 × 10 mm dishes (Corning). 1 × 10$^3$ BM cells were plated in triplicate and cultured at 37 °C. Colonies were counted after 7 days, prior to passage.

**Quantitative real-time PCR.** RNA was converted to cDNA using the Maxima H Minus First Strand cDNA Synthesis kit (Thermo Fisher) and quantitative real-time PCR (qRT-PCR) was performed using SYBR Green (Thermo Fisher) according to the manufacturer's instructions. Cdx2 (5′-CAAGGACGTGAGCATGTATCC-3′, 5′-GTAACCACCGTAGTCCGGGTA-3′) messenger RNA levels were determined from sorted PB and BM cells normalized to GAPDH (5′-AGGTCGGTGTGAACG GATTTG-3′, 5′-TGTAGACCATGTAGTTGAGGTCA-3′).

**Western blot.** Immunoblot for Scl-CreER$^T$ and Scl:Cdx2 cells was performed using mouse anti-CDX2 monoclonal antibody (CDX-88; Abcam) (1:1000) and mouse anti-β-actin (BD Biosciences) (1:5000). Immunoblot for rabbit anti-Flag immunoprecipitation of Ba/F3 cells transduced with pMIG-Cdx2-Flag was performed using mouse anti-Flag tag monoclonal antibody (8146, Cell Signaling) (1:1000).

**Reporting summary**. Further information on research design is available in the Nature Research Reporting Summary linked to this article.

## Data availability

The RNA-Seq datasets generated and analysed during the current study are available in the GEO database (https://www.ncbi.nlm.nih.gov/geo) under the SuperSeries accession number GSE133829 (GSE133679, BM LKS four weeks after tamoxifen treatment; GSE133680, Cdx2-mediated AML treated with Azacitidine or vehicle; GSE133828, Cdx2-mediated acute leukemia BM LKS). Publicly available datasets published by Lara-Astiaso et al.[43] was obtained from the GEO database, accession numbers GSE60101 (RNA-Seq) and GSE59992 (ATAC-Seq). Publicly available dataset for Cebpa ChIP-Seq on mouse GMP was obtained from GEO database, accession number GSM1187163.

The ChIP-Seq dataset generated during this study on Cdx2-FLAG-transduced mouse BM is available at the accession number GSE146598. The ATAC-Seq dataset performed on BM LKS four weeks after tamoxifen treatment can be accessed here: https://genome.ucsc.edu/s/JasminS/VU_2019_CDX2_ATAC. Whole exome sequencing (WES) datasets performed on Cdx2 mouse BM are available at the Sequence Read Archive (SRA) with accession number PRJNA552223. Whole exome sequencing data for Supplementary Fig. 3a and Supplementary Table 1 can be found in Supplementary Data 1. RNA-Seq data for Fig. 5a–h and Supplementary Fig. 5d–f can be found in Supplementary Data 2. RNA-Seq data for Fig. 7j–m, Supplementary Fig. 5f and Supplementary Fig. 7d–g can be found in Supplementary Data 5. All other data supporting the findings of this study are available within the article and its supplementary information files and from the corresponding authors upon reasonable request.

## Code availability

No custom code was generated during this study.

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

## Acknowledgements

We would like to thank members of the Lane Lab, Geoff Hill, and Kyle MacBeth and Daniel Menezes from Celgene for helpful discussions and input, and Michael Milsom and Ruzhica Bogeska for technical assistance with flow sorting. We are grateful for the expert assistance of the QIMR Berghofer Flow Core, Animal House and Sequencing Core. This work was funded by NHMRC Project Grant 1098470, a philanthropic donation (Gordon and Jessie Gilmour Trust) and a research grant in aid from Celgene. Celgene provided 5-azacitidine for the mouse experiments. S.W.L. is a CSL Centenary Fellow. T.V. received a Leukaemia Foundation of Australia PhD Scholarship.

## Author contributions

T.V. and S.W.L. conceptualized and designed experiments. T.V., J.S., and S.W.L. analyzed and interpreted data, and wrote the manuscript. T.V., A.H.P., M.B., A.S., V.L., L.C., G.P., C.B., S.J., J.G., and G.M. performed experiments and data collection. J.S., G.M., A.M.C., P.M., and J.P.M. performed bioinformatical analyses. A.P., C.R.W., F.H.H., N.C., S.G., J.P.M., S.F., C.S., and S.W.L. provided critical reagents and data interpretation, and/or intellectual input and supervision of the study. All authors contributed to and edited the manuscript.

## Competing interests

We have the following disclosures that have been included in the manuscript as potential for conflict of interest. This work was funded by the National Health and Medical Research Fund of Australia and the Leukaemia Foundation of Australia. Celgene provided azacitidine and additional research funding. S.W.L. has participated in an advisory board for Celgene.
