## [Peer Review File · Nature Communications]

Reviewers' comments:

Reviewer #1 (Expertise: AML/MDS models, Remarks to the Author):

In this manuscript Vu and Colleagues present detailed phenotypic, cellular and molecular characterization of a novel mouse strain that models the evolution of MDS to AML. This model expresses the orphan homeobox transcription factor Cdx2 from the ROSA26 locus and combines this with a T2A linked mcherry construct to use as a reporter. The mice develop a spectrum of haematological diseases (MPN, MDS, AML, B-ALL and T-ALL), and show evolution from MDS to AML associated with the acquisition of additional driver mutations. Molecularly Cdx2 expression alters gene expression programmes associated with a major acquisition of chromatin accessibility. Surprisingly, given the published literature this is not associated with an overexpression of Hoxa and Hoxb cluster genes. Finally the authors use this novel model to probe the mechanisms and optimal administration of 5-azacytidine (AZA), demonstrating evidence that lower dose AZA demonstrates greater efficacy.

Overall this is an interesting study; the mouse is very well characterized on multiple levels and is likely to be a useful model to study leukemic progression of MDS. Moreover, the model may also be useful to study therapeutics for MDS, AML and the progression between these states and this study has obvious clinical implications. I have the following questions, concerns and issues that should be addressed.

1. Some estimation of the Cdx2 transcript and protein level and how this compares with the retroviral models and human AML patient samples would be very helpful in terms of demonstrating that this level is physiological (vs human AML) in the new model and potentially for comparisons between the transgenic and viral models, to account for the differences seen in Hox gene regulation.
2. Although the Kaplan Meier for all of the animals is shown in Fig 3 (to include the +Flt3-ITD combination) , I think that this also needs to be shown in Fig2. The latency of the disease can be worked out from 2C but this is quite difficult to do. As most of the following studies utilize mice numbers 2259, 252, 882 and 472, these should be identified on the KM and current Figure 2C
3. Did any AML cases demonstrate an MPN prodromal phase?
4. Is the acute leukaemia phenotype stable in the secondary transplants? Figure 2i for mouse number 882 would suggest so for the T-cell disease, but what about B-ALL and AML?
5. Could the authors clarify the number of mice on which WES was performed - is this only 3? Also, can the authors clarify the source of the germline control material please? Was this material harvested from the non-cherry positive cells from the same donor mice (the ideal control) or from the non-cherry cells of the transplant recipients?

6. Apart from it being usually a strongly synergistic oncogene in AML induction, why was Flt3-ITD chosen as a cooperating mutation for the studies in Figure 3. Is CDX2 expression higher in FLT3-ITD positive rather than FLT3 WT human AML cases?
7. Aspects of the Cdx2 aberrant/overexpression phenotype are reminiscent of a signaling abnormality (proliferation, differentiation, loss of HSC function, increased cell cycle activity, MPN phenotype, synergism with Flt3-ITD to generate a more aggressive fatal MPN rather than AML) rather than an oncogenic transcription factor. Do the authors see any alteration of signaling in the early Cdx2 mice?
8. Figure 6 – panel a – For the shared peaks at both promoters and distal enhancer peaks the number given is 6679. I think that this is perhaps an error. Could the authors please check this. For pnael e, the subtraction peak in green should be scaled the same as the blue and red Scl:Cdx2 and Scl-Cre peaks at max 1.2.
9. Regarding the role of haematopoietic TF such as Cebpa, b,d and e the link would be less speculative if the authors could use haematopoietic ChIP-Seq data sets from the published domain to compare and demonstrate an overlap. These are certainly available in HPC-7 cell line and probably also in Lin- or LSK cells.
10. The clinical studies are potentially very interesting, particularly in light of the increasing availability of oral azacytidine (CC-486). How do the LD and HD regimens used in this study compare to human dosing?
11. The clinical studies may suggest that tp53/TP53 needs to be intact for the therapeutic effect. Is there any suggestion from in vitro work that this is the case; for example that tp53 WT case 2259 is more sensitive than tp53 mutated case 472?
12. In figure 7m what are Scl-cre 1o and 2o
13. The statement in the discussion that “When Cdx2 was expressed in HSCs, the secondary mutation appeared to dictate the lineage specificity of the leukemia” is currently too strong. It is based on 3 cases and in light of this I think that the authors should tone this down.

Reviewer #2 (expertise: AML/MDS models, therapy, Remarks to the Author):

Vu et al. have used a novel transgenic mouse line to characterize the effects of Cdx2 expression on hematopoiesis and leukemogenesis. The rationale is that Cdx2 is overexpressed in many human leukemias, including ALL and AML, yet prior studies into its function have been restricted to retroviral models. The authors have shown that Cdx2 over-expression depletes the HSC pool and increases GMP frequency. A subset of transgenic mice developed AML (or ALL) that could be accelerated by FLT3-ITD. Many mice also had pre-leukemic MDS-like disease. Progression to frank leukemia required expression in an uncommitted progenitor, as LyzM-Cre; Cdx2 mice only developed a myeloproliferative neoplasm. The leukemias were sensitive to azacytidine.

Among the more surprising findings was evidence showing that Cdx2 suppresses Hox gene expression, and the Hox genes (as well as other HSC-enriched genes) are not re-activated in fully transformed AML cells. Gene expression and ATAC-seq profiling show that Cdx2 promotes myeloid commitment.

This is an interesting, clearly-written manuscript that should extend our understanding of how Cdx2 helps drive leukemogenesis. There are some significant issues – particularly with interpretation of some of the data – that should be addressed.

Major issues:

1. In several places, the authors refer to the HSC as the cell of origin for the AML, and/or they claim that HSCs are transitioning to LSCs. However, the data cannot resolve whether the cell of origin is an HSC or a more committed HPC. It seems equally, if not more plausible that a clone within the MPP/HPC population (i.e. a CD48+LSK) could acquire mutations and give rise to the leukemias. The CD48+ population is preserved in the Cdx2 mice whereas the phenotypic HSCs are almost gone. If the authors want to directly test their argument, they could perform the experiment with Flk2-Cre mice, but a simpler solution would be to rephrase the argument.

2. Along the same lines, it would be helpful to know more about the mechanism of transformation. The authors show, compellingly, that Hox genes, Mecom and other self-renewal genes are inactivated in pre-leukemic HPCs, and they are not ectopically re-activated in AML cells. So how do the leukemia cells avoid terminal differentiation? One possibility is that effectors of myeloid commitment (Runx1, PU.1, CEBPA, etc.) are selectively downregulated in the AML cells. A western blot showing expression of these proteins in normal, Cdx2 and Cdx2-AML HPCs and GMPs might give some more insight into how these effectors fluctuate in pre-leukemic progenitors and AML cells. The mutations identified by exome sequencing don't really speak to this issue.

3. The ATAC-seq data do not establish a direct role for CDX2 in activating myeloid enhancers or repressing Hox gene enhancers. All of the epigenetic reprogramming that the authors describe could be an indirect consequence of premature myeloid commitment within the LSK population. Enrichment of CDX2 sites in the differential peaks is consistent with a direct role for CDX2, but it does not prove the interaction.

This is the most interesting and mechanistic segment of the paper. The impact of the story will improve considerably if they can establish whether CDX2 binds a subset of these enhancer elements, especially the elements near the Hox genes. I recognize that the technical limitations of ChIP-seq in HPCs, as well as the limitations of cell line experiments. Nevertheless, the authors should try to test whether CDX2 interacts with putative elements in some experimental context (e.g. transduce 32D or HPC5 cells with a Flag-CDX2 for ChIP-seq or ChIP-qPCR). Without these data, the evidence for direct repression of the Hox loci, or direct activation of myeloid enhancers, is less compelling.

Minor:

1. Please label the tracks more clearly in figure 6e. The labels from the UCSC browser are difficult to read and interpret.

2. The authors should show absolute numbers and not just frequencies for the populations in Fig 1e-i. It is not clear whether Cdx2 expression leads to GMP expansion or whether all progenitor populations become depleted. Is the marrow hypercellular or hypocellular or normocellular?

3. Could the authors please clarify whether Cdx2 is expressed at any point during normal adult hematopoiesis? Does expression increase with myeloid commitment and inversely correlate with Hox gene expression in Supplementary Fig 5E, or is it an exclusively embryonic gene that gets reactivated in AML cells?

Reviewer #3 (Expertise: RNAseq, ATACseq, leukemia, Remarks to the Author):

Vu et al characterized the role of CDX2, a caudal-related transcriptional regulator of homeobox genes, in hematopoiesis and leukemia in mice by establishing a Cdx2 conditional knock-in (KI)

overexpression mouse model. They found that hematopoietic selective (by Scl-CreERT) tamoxifen-inducible Cdx2 overexpression impaired HSC function and the differentiation of myeloid lineages, resulting in the development of lethal myeloid and lymphoid malignancies. Cdx2 overexpression restricted to the myeloid progenitors and differentiated myeloid cells promoted the development of MPNs but not secondary leukemia. Combining Cdx2 overexpression and Flt3-ITD did not accelerate AML development. By ATAC-seq and RNA-seq analyses of LSK cells with or without Cdx2 overexpression, the authors observed changes in gene expression and chromatin accessibility associated with HSC self-renewal, lineage differentiation and cell proliferation etc. Interestingly, Cdx2 overexpression in HSPCs resulted in significant downregulation of HoxA and HoxB genes in contrast to previously reported roles of Cdx2 in regulating Hox gene. Finally, they showed that low-dose azacitidine treatment displayed greater efficacy associated with specific hypomethylation in Cdx2 overexpression-induced myeloid malignancies.

The main strengths of the manuscript are the generation of a new mouse model for Cdx2 overexpression, and the phenotypic analysis of various hematopoietic and disease phenotypes associated with Cdx2 overexpression in mouse models. The mouse genetic experiments were appropriately designed and the results were carefully analyzed. Therefore, this study provides a nice addition to the existing literature on the function of Cdx2 in HSC function and leukemia (Scholl et al 2007 JCI 117:1037; Rawat et al., 2004 PNAS 101:817; Rawat et al., 2007 Blood 111:309; etc). The overall findings also support an important role of Cdx2 in the development of hematopoietic malignancies in vivo. The authors made other findings, including the effects of Cdx2 overexpression on LT-HSCs and lineage differentiation, and the unexpected downregulation of Hox genes, that increased our understanding of this important transcription factor in hematopoiesis and leukemia. However, the current study falls short in providing sufficient mechanistic insights into how Cdx2 regulates target genes and chromatin accessibility to affect HSC differentiation, leukemia progression and transformation, and how Cdx2 overexpression promoted development of both myeloid and lymphoid malignancies. The lack of Cdx2 ChIP-seq studies also makes it difficult to identify Cdx2 direct gene targets vs the indirect ATAC-seq and RNA-seq changes in HSPCs. Some important experiments and/or analysis are also lacking, such as CDX2 expression correlation with human myeloid and lymphoid malignancies, the regulation of Hox genes by Cdx2, and the detailed phenotypic and molecular analyses of lymphoid hematopoiesis. Therefore, although the current study provides some interesting findings and the experiments were carefully executed, the overall conclusions were not developed in sufficient depth in light of previous studies. There are several important remaining questions that need to be addressed, as detailed below.

Major points:

1. Since the Cdx2 KI overexpression by Scl-CreERT resulted in the development of both myeloid and lymphoid malignancies (Fig. 2), it will be important to analyze the effect of Cdx2 overexpression on normal lymphoid development (e.g. both T and B lineages and their progenitors).

2. Detailed phenotypic analysis of the Cdx2 overexpression-induced lymphoid malignancies should be included. Similarly, detailed molecular analysis of gene expression and chromatin changes in the affected lymphoid populations should be included and compared to that in the HSPCs or myeloid cells. This line of investigation will reveal shared and unique target genes and associated cellular pathways responsible for Cdx2-mediated leukemia transformation in different hematopoietic lineages.

3. Cdx2 overexpression significantly impaired LT-HSCs (Fig. 1H-I). Given the significant effects, it is surprising that the authors did not analyze other HSPC populations such as ST-HSCs and MPPs. It would be important to determine the molecular causes of LT-HSC exhaustion (e.g. impaired quiescence, proliferation, and/or differentiation).

4. Cdx2 ChIP-seq should be performed and compared with ATAC-seq and RNA-seq datasets to define gene targets that may be directly regulated by Cdx2. The authors stated that ChIP-seq was not possible using rare HSPC populations. However, there are many modified ChIP-seq protocols that can work with thousands, hundreds of cells or even single cells, such as ChIPmentation and CUT&RUN (Schmidl et al., Nature Methods 12:963; Skene et al., Nature Protocols 13:1006; Hainer et al., Cell 177:1319).

5. In Fig. 6E and Supplementary Fig. 5D, the authors included results that Cdx2 overexpression increased chromatin accessibility at predicted Cdx2 binding sites at HoxA and HoxB gene clusters, and Cdx2 associates with the same binding sites in other cell types (ES and endoderm) by ChIP-seq. These results suggest that Cdx2 may function to repress HoxA and HoxB gene expression in HSPCs in contrast to its activating roles in other cellular contexts. If validated these results could be highly significant that also explain the observed downregulation of Hox genes in Cdx2 overexpressing HSPCs. Additional studies would be necessary to follow up on these interesting results.

6. Given the observed phenotypes in both myeloid and lymphoid malignancies upon Cdx2 overexpression. It is important to also provide corroborative analyses of CDX2 expression in human myeloid and lymphoid malignancies. Is CDX2 overexpressed in human myeloid and lymphoid malignancies and correlated with poor survival and therapeutic responses? Is there difference in CDX2 expression with specific genetic lesions in myeloid and lymphoid malignancies? Does CDX2 expression positively or negatively correlate with HOX gene expression in different contexts?

Minor points:

1. The use of Flt3-ITD mouse model in combination of Cdx2 overexpression was not justified. Is CDX2 overexpression more commonly seen in human AML with FLT3-ITD mutations?

2. Typo page 5 line 4, 'decreased' should be 'decreases'.

Thank you for the opportunity to submit a revised manuscript. We include a point by point response to the Reviewers' constructive feedback, together with new data in each case that helps to address the concerns raised. We believe that the additional data and revisions have substantially improved the manuscript and are grateful for the useful suggestions provided during the review process. We hope that this revised manuscript will now be suitable for publication as an original research article in Nature Communications.

Sincerely,
Steven Lane, on behalf of the co-authors.

Reviewer #1 (Expertise: AML/MDS models, Remarks to the Author):

1. Some estimation of the *Cdx2* transcript and protein level and how this compares with the retroviral models and human AML patient samples would be very helpful in terms of demonstrating that this level is physiological (vs human AML) in the new model and potentially for comparisons between the transgenic and viral models, to account for the differences seen in *Hox* gene regulation.

Thank you for this interesting question. Using qPCR, we compared retroviral expression of *Cdx2* (MSCV-IRES-GFP-*Cdx2*) in *Scl*-CreER^T GFP-sorted BM cells with transgenic expression of *Cdx2* in *Scl*:*Cdx2* mCherry-sorted BM cells. We observed that retroviral CDX2 overexpression resulted in higher levels than the transgenic model. This new figure replaces Supplementary Figure 1C with the following text also substituted.

Main text:

To evaluate *Cdx2* expression differences between previously published retroviral models¹ and *Scl*:*Cdx2* transgenic cells, we transduced *Scl*-CreER^T BM cells with MSCV-IRES-GFP (MIG)-*Cdx2* and MIG-Empty retrovirus. Retroviral CDX2 overexpression resulted in approximately 900-fold higher CDX2 expression than *Scl*:*Cdx2* transgenic cells, potentially accounting for phenotypic differences (Supplementary Figure 1D).

Figure legend.

(c) qPCR of bone marrow (BM) cells from *Scl*-Cre and *Scl*:*Cdx2* after tamoxifen induction. *Scl*-Cre cells were lineage-depleted and transduced with MSCV-IRES-GFP (MIG)-*Cdx2*¹ or MIG-Empty retrovirus and sorted for GFP. *Scl*:*Cdx2* samples were sorted for mCherry to show specific *Cdx2* expression relative to Actin and GAPDH housekeeping genes.

2. Although the Kaplan Meier for all of the animals is shown in Fig 3 (to include the +Flt3-ITD combination) , I think that this also needs to be shown in Fig2. The latency of the disease can be worked out from 2C but this is quite difficult to do. As most of the following studies utilize mice numbers 2259, 252, 882 and 472, these should be identified on the KM and current Figure 2C

Thank you for this important suggestion. We have updated the figures and figure legends to include:

1. Kaplan-Meier for *Scl:Cdx2* vs. *Scl-CreERT* in Supplementary Figure 2A.
2. Annotated acute leukaemia mouse numbers in Figure 2C and Supp Fig 2A.

The updated manuscript text now reads:

“*Scl:Cdx2* mice had a median survival of 43 weeks while no disease was seen in *Scl-CreER^T* controls (Supplementary Figure 2A)”

3. Did any AML cases demonstrate an MPN prodromal phase?

Thanks for this interesting question. We did not observe an MPN prodromal phase prior to AML. AML was only preceded by a MDS phase (low blood counts and dysplasia).

4. Is the acute leukaemia phenotype stable in the secondary transplants? Figure 2i for mouse number 882 would suggest so for the T-cell disease, but what about B-ALL and AML?

Yes, all acute leukaemia phenotypes were transferred and stable in the transplant models. The manuscript text has been updated to clarify this:

“The leukemias were transplantable as irradiated recipient mice phenocopied the primary donor in all cases (example in Figure 2I)”

5. Could the authors clarify the number of mice on which WES was performed - is this only 3? Also, can the authors clarify the source of the germline control material please? Was this material harvested from the non-cherry positive cells from the same donor mice (the ideal control) or from the non-cherry cells of the transplant recipients?

Thank you for pointing out this lack of clarity. The text has now been modified to clarify these points (in both Results and Methods sections):

“we performed whole exome sequencing (WES) of three AL samples and one MPN sample. WES was performed on genomic DNA of CD45.2-sorted cells (ie. donor cells) from transplanted leukemic mice. Tumor samples were sourced from mCherry-positive donor cells and compared to germline samples that were mCherry-negative donor cells.”

“BM cells from *Scl-CreER^T*-transplanted and *Scl:Cdx2* secondary leukemia mice were sorted for CD45.2 to isolate donor cells, and then mCherry-positive and mCherry-negative fractions.”

6. Apart from it being usually a strongly synergistic oncogene in AML induction, why was Flt3-ITD chosen as a cooperating mutation for the studies in Figure 3. Is CDX2 expression higher in FLT3-ITD positive rather than FLT3 WT human AML cases?

The Reviewer is correct that FLT3-ITD is often synergistic with transcription factor alterations in AML. This was a primary consideration but we are grateful for the opportunity to explain further the rationale for this strategy.

We measured CDX2 and FLT3 mRNA expression by RT-qPCR in 60 AML patient blasts and found significant co-expression ($p=0.0017$, Reviewer Figure 1). These results were validated using the publicly available Beat AML dataset consisting of 562 AML patients (Reviewer Figure 2a). Finally, using the same Beat AML cohort, we found significantly increased CDX2 expression in FLT3-ITD-positive patients compared to those that were FLT3-ITD-negative (Figure 2b). Reference to this has been added to the manuscript text as follows:

“Using the Beat AML trial cohort (Tyner *et. al.* 2018), we found significant co-expression of CDX2 and FLT3 in AML patients, as well as increased CDX2 expression in FLT3-ITD-positive samples compared to FLT3-ITD-negative samples (data not shown). We therefore tested whether *Scf:Cdx2* mice would accelerate development of AML when crossed with mice harboring *Flt3-ITD*, a common oncogene in AML”

Reviewer Figure 1. CDX2 and FLT3 mRNA relative to PBGD housekeeping control.

Reviewer Figure 2. Beat AML patient cohort.

a) CDX2 and FLT3 expression.

compared to FLT3-ITD-negative samples.

b) Increased CDX2 read counts in FLT3-ITD-positive AML samples

7. Aspects of the Cdx2 aberrant/overexpression phenotype are reminiscent of a signaling abnormality (proliferation, differentiation, loss of HSC function, increased cell cycle activity, MPN phenotype, synergism with Flt3-ITD to generate a more aggressive fatal MPN rather than AML) rather than an oncogenic transcription factor. Do the authors see any alteration of signaling in the early Cdx2 mice?

This is an interesting point. We did not see evidence suggesting activation of classical signalling pathways in the early Cdx2 mice that would contribute to a proliferative phenotype using gene expression pathway analysis. In fact, we observed that certain signalling pathways such as interferon signalling and TNF α via NF κ B, were suppressed in these mice. We believe that this reflects the block of differentiation into mature, cytokine-producing cells. These results are provided here as Reviewer Figure 3.

Reviewer Figure 3.

GSEA for Scl-Cre vs Cdx2

NAME	ES	NES	NOM p-val	FDR q-val
HECKER_IFNB1_TARGETS	-0.7372	-2.15828	0	0
REACTOME_INTERFERON_ALPHA_BETA_SIGNALING	-0.70965	-2.08774	0	1.39E-04
MOSERLE_IFNA_RESPONSE	-0.84508	-2.03182	0	4.82E-04
BROWNE_INTERFERON_RESPONSIVE_GENES	-0.64382	-1.93053	0	0.005795
TAVOR_CEBPA_TARGETS_DN	-0.75997	-1.91166	0	0.007168
LIANG_SILENCED_BY_METHYLATION_2	-0.72303	-1.8511	0	0.018876
DAUER_STAT3_TARGETS_DN	-0.67548	-1.83623	0.001397	0.022755
TIAN_TNF_SIGNALING_NOT_VIA_NFKB	-0.72388	-1.77248	0.001513	0.043834
NAGASHIMA_EGF_SIGNALING_UP	-0.63328	-1.76354	0	0.04769

GSEA for Cdx2 vs Scl-Cre

NAME	ES	NES	NOM p-val	FDR q-val
ZHENG_IL22_SIGNALING_UP	0.812261	2.387883	0	0
CROONQUIST_IL6_DEPRIVATION_DN	0.517731	2.070075	0	0.001915

8. Figure 6 – panel a – For the shared peaks at both promoters and distal enhancer peaks the number given is 6679. I think that this is perhaps an error. Could the authors please check this. For pnael e, the subtraction peak in green should be scaled the same as the blue and red Scl:Cdx2 and Scl-Cre peaks at max 1.2.

We apologise for this error and thank the Reviewer for pointing this out. The correct value is 12592 (67% shared). Likewise, we have updated the figures in panel 6e so that the scale of the subtraction peak is the same as the red and blue peaks.

9. Regarding the role of haematopoietic TF such as Cebpa, b,d and e the link would be less speculative if the authors could use haematopoietic ChIP-Seq data sets from the published domain to compare and demonstrate an overlap. These are certainly available in HPC-7 cell line and probably also in Lin- or LSK cells.

Thank you for this excellent suggestion. We note that the binding of Cdx2 was a key query of all reviewers and hence, we have devoted much of this revision to answering this important question.

Firstly, as suggested, we have compared our ATAC-Seq data to the publicly available CEBPA ChIP-Seq dataset GSM1187163 performed in GMPs and found a significant overlap in peaks in Scl:Cdx2 BM but not Scl-Cre control samples (included in manuscript Figure 6C and Supp Figure 6D).

Furthermore, we have now performed ChIP-Seq on Cdx2 in hematopoietic cells to validate these findings. We transduced lineage-depleted mouse BM with retroviral Cdx2-FLAG construct. Using this approach, we were able to identify specific Cdx2 binding and strong motif enrichment for Cdx2 (Supplementary Figure 6A for immunoprecipitation validation and Figure 6B for motif enrichment). These data were compared to ATAC-seq gained peaks, and we have confirmed Cdx2-FLAG binding at the gained peaks on ATAC-sequencing and show strong overlap with CEBPA/B regulatory regions (Figure 6C).

For ease of review, the data are presented here.

Reviewer Figure 4, Paper Figure 6B, C and Supplementary Figure 6D.

Motif enrichment analysis of Cdx2-FLAG ChIP-Seq samples showing Cdx2 is centrally enriched at promoter and distal elements

Heatmap of top 1000 gained peaks at distal elements in Cdx2-FLAG ChIP-Seq that overlap with Scl:Cdx2 or Scl-cre ATAC-Seq

Distal element peak overlap with CEBPA GMP ChIP-Seq. Presented is the number of *Scl-Cre* and *Scl:Cdx2* specific distal peaks overlapping with CEBPA ChIP-Seq peaks from GMPs.

Distal element peak overlap

Peaks	CEBPa GMP peak overlap GSM1187163	No overlap
Scl-Cre	92 (10.1%)	813 (89.9%)
Scl:Cdx2	5738 (39.6%)	8746 (60.4%)

Chi-Square test $p < 0.0001$

10. The clinical studies are potentially very interesting, particularly in light of the increasing availability of oral azacytidine (CC-486). How do the LD and HD regimens used in this study compare to human dosing?

Thank you for this question. This finding is key to the clinical impact of this paper and these preclinical data have been included in the upcoming registration application by Celgene for CC-486 to the FDA. The Reviewer correctly points out that the LD extended duration regimen is based on CC-486 regimens, as per clinical trials sponsored by Celgene (clinicaltrials.gov identifier: NCT00528983 and cited as Laille *et al.* 2015: ref #46, Garcia-Manero *et al.* 2016: ref #47). We have modified text in the discussion section to highlight these studies.

“Using dosing schedules comparable to CC-486 oral Aza regimens used in human clinical trials^{46,47}, Aza appeared to be more effective and more specific for hypomethylation when administered in a low-dose, continuous regimen compared to high-dose intermittent treatment.”

11. The clinical studies may suggest that *tp53/TP53* needs to be intact for the

therapeutic effect. Is there any suggestion from in vitro work that this is the case; for example that tp53 WT case 2259 is more sensitive than tp53 mutated case 472?

Thank you for this excellent suggestion. We have performed in vivo studies with azacitidine on the tp53-null Cdx2 leukemia strain 472. In comparison to other Cdx2 leukemia strains (252, 882, 2259), azacitidine did not improve survival of tp53-mutated 472 mice (Reviewer Figure 5).

Reviewer Figure 5.

These are provocative preclinical data. Before submitting this work for publication we would like to confirm these results and perform further characterization of the role of TP53 in azacitidine response, however we believe that this work is beyond the scope of this publication.

12. In figure 7m what are Scl-cre 1o and 2o

Thank you for allowing us to clarify this. We have amended the figure to specify Scl-cre primary mice and Scl-cre secondary (transplanted) mice.

13. The statement in the discussion that “When Cdx2 was expressed in HSCs, the secondary mutation appeared to dictate the lineage specificity of the leukemia” is currently too strong. It is based on 3 cases and in light of this I think that the authors should tone this down.

Thank you for this suggestion. We have modified the discussion to now state: “When *Cdx2* was expressed in HSPCs, mice showed a propensity to develop secondary mutations followed by the development of a range of acute leukemias of varying lineages.”

Reviewer #2 (expertise: AML/MDS models, therapy, Remarks to the Author):

Major issues:

1. In several places, the authors refer to the HSC as the cell of origin for the AML, and/or they claim that HSCs are transitioning to LSCs. However, the data cannot resolve whether the cell of origin is an HSC or a more committed HPC. It seems equally, if not more plausible that a clone within the MPP/HPC population (i.e. a

CD48+LSK) could acquire mutations and gives rise to the leukemias. The CD48+ population is preserved in the Cdx2 mice whereas the phenotypic HSCs are almost gone. If the authors want to directly test their argument, they could perform the experiment with Flk2-Cre mice, but a simpler solution would be to rephrase the argument.

Thank you for the chance to make this important clarification. As suggested, we have rephrased the argument to reflect the possibility that an intermediate MPP population may give rise to the leukemias. We have changed all references to HSC to HSPC, including in the title of the manuscript.

2. Along the same lines, it would be helpful to know more about the mechanism of transformation. The authors show, compellingly, that Hox genes, Mecom and other self-renewal genes are inactivated in pre-leukemic HPCs, and they are not ectopically re-activated in AML cells. So how do the leukemia cells avoid terminal differentiation? One possibility is that effectors of myeloid commitment (Runx1, PU.1, CEBPA, etc.) are selectively downregulated in the AML cells. A western blot showing expression of these proteins in normal, Cdx2 and Cdx2-AML HPCs and GMPs might give some more insight into how these effectors fluctuate in pre-leukemic progenitors and AML cells. The mutations identified by exome sequencing don't really speak to this issue.

This is an excellent point regarding the regulation of myeloid transcription factors and their effect on AML progression. In early Cdx2 LSK cells (which represent pre-leukemic HSPC) we observed upregulation of *Spi1* and *Cebp* genes (*Cebpa*, *Cebpb*, *Cebpd* and *Cebpe*) and downregulation of *Runx1* (Reviewer Figure 6). This is consistent with hematopoietic progenitor differentiation toward GMP and mature myeloid cells. In contrast, transformed Cdx2 LSK cells from acute leukemic mice show similar or decreased levels of *Spi1* and *Cebp* gene transcripts compared to control cells. This suggests, as submitted by the Reviewer, that AML cells appear to selectively downregulate effectors of myeloid commitment during AML transformation. We also observe that in AML cells responding to azacitidine, the same myeloid transcription factors are subsequently upregulated, suggesting a restoration of pro-differentiation pathways in the context of hypomethylating agents.

Reviewer Figure 6.

A) Heat map showing relative gene expression levels of myeloid transcription factors in wild-type (Scl-Cre), pre-leukemic Cdx2 and transformed Cdx2 BM cells. Also shown are gene expression levels in leukemic mice treated with low exposure or high exposure azacitidine

- Scl-cre primary mice
- Scl-cre secondary (transplanted)
- Scl:Cdx2 - Leukemia
- Veh: Scl:Cdx2 - Leukemia
- Aza HE-LD
- Aza LE-ED
- Scl:Cdx2 - Pre-leukemia

B) Heat map showing relative expression levels of Cebp genes in distinct hematopoietic populations (Lara-Astiaso *et. al.* 2014)

To incorporate this interesting data into the paper, we have included the below text and figure into the manuscript.

Supplementary Figure 5F.

RNA-Seq data showing expression of transcription factor expression in pre-leukemic Cdx2 HSCs (4 weeks after tamoxifen) and established leukemias.

“In support of this, RNA-Seq also showed upregulation of *Sp1* and *Cebp* family genes and downregulation of *Runx1* in Scl:Cdx2 LKS+ (representing pre-leukemic HSPC) in keeping with myeloid differentiation. Interestingly, transformed Scl:Cdx2 LKS+ BM cells from acute leukemic mice showed similar or decreased levels of *Sp1* and *Cebp* gene transcripts compared to control cells (Supplementary Figure 5F), suggesting these leukemia cells downregulate effectors of myeloid commitment as a mechanism of transformation.”

3. The ATAC-seq data do not establish a direct role for CDX2 in activating myeloid enhancers or repressing Hox gene enhancers. All of the epigenetic reprogramming that the authors describe could be an indirect consequence of premature myeloid commitment within the LSK population. Enrichment of CDX2 sites in the differential peaks is consistent with a direct role for CDX2, but it does not prove the interaction.

This is the most interesting and mechanistic segment of the paper. The impact of the story will improve considerably if they can establish whether CDX2 binds a subset of these enhancer elements, especially the elements near the Hox genes. I recognize that the technical limitations of ChIP-seq in HPCs, as well as the limitations of cell line experiments. Nevertheless, the authors should try to test whether CDX2 interacts with putative elements in some experimental context (e.g. transduce 32D or HPC5 cells with a Flag-CDX2 for ChIP-seq or ChIP-qPCR). Without these data, the evidence for direct repression of the Hox loci, or direct activation of myeloid enhancers, is less compelling.

Thank you for this suggestion. We have transduced lineage-negative mouse BM with Cdx2-FLAG retrovirus and performed ChIP-Seq with anti-FLAG antibody. We confirmed specific pulldown on immunoprecipitation (Supp Fig 6A) and confirmed strong overlap between Cdx2 binding and differential ATACseq peaks (Supp Fig 6B). There was enrichment for putative Cdx2 binding (Fig 6B).

We have been able to use this strategy to show that Cdx2 does indeed bind regulatory regions for myeloid transcription regulators (CEBPa, Figure 6C) and Hox gene enhancers (Fig 6E), correlating our ATAC-Seq data in primary Cdx2 transformed cells. Thus, we suggest that Cdx2 directly interacts with myeloid and Hox loci to regulate mature myeloid cell differentiation and suppress HSPC self-renewal. These data are included in the manuscript and are also shown above in response to Reviewer 1 (Reviewer Figure 4).

Minor:

1. Please label the tracks more clearly in figure 6e. The labels from the UCSC browser are difficult to read and interpret.

We apologise. We have labelled the tracks to be more clear. ChIP-seq tracks are now included showing Cdx2 binding.

2. The authors should show absolute numbers and not just frequencies for the populations in Fig 1e-i. It is not clear whether Cdx2 expression leads to GMP expansion or whether all progenitor populations become depleted. Is the marrow hypercellular or hypocellular or normocellular?

Thank you. We observed largely normocellular bone marrow in Cdx2 mice, and with this a slight decrease in CMP and MEP, however GMP absolute numbers were unchanged. We will include these graphs in the manuscript and amend the text as follows:

“Decreased absolute CMP and MEP cell counts were also observed in Scl:Cdx2 BM, despite Scl:Cdx2 BM being normocellular (Supplementary Figure 11-L).”

Supplementary Figure 11-L reproduced here for convenience.

3. Could the authors please clarify whether Cdx2 is expressed at any point during normal adult hematopoiesis? Does expression increase with myeloid commitment and inversely correlate with Hox gene expression in Supplementary Fig 5E, or is it an exclusively embryonic gene that gets reactivated in AML cells?

Thank you. In both mouse and human data, we find that Cdx2 is not expressed in normal HSPC or committed hematopoietic cells (Reviewer Figure 7). Hence, our analysis supports the theory that CDX2 is an embryonic gene that is reactivated in acute leukemia.

Reviewer Figure 7.

Figure 1 taken from Scholl *et al.* (2007) showing absence of CDX2 expression in normal human BM cells.

Figure 1

CDX2 expression in AML. CDX2 mRNA levels were measured by RQ-PCR in 170 AML patients from different cytogenetic subgroups, as well as in BMNCs ($n = 10$), CD34⁺ cells ($n = 3$), HSCs ($n = 3$), CMPs ($n = 3$), GMPs ($n = 3$), and MEPs ($n = 3$) from normal individuals. Circles indicate patients with genomic amplification of the CDX2 locus, as assessed by aCGH and FISH (25). Bars indicate median values.

Reviewer #3 (Expertise: RNAseq, ATACseq, leukemia, Remarks to the Author):

However, the current study falls short in providing sufficient mechanistic insights into how Cdx2 regulates target genes and chromatin accessibility to affect HSC differentiation, leukemia progression and transformation, and how Cdx2

overexpression promoted development of both myeloid and lymphoid malignancies. The lack of Cdx2 ChIP-seq studies also makes it difficult to identify Cdx2 direct gene targets vs the indirect ATAC-seq and RNA-seq changes in HSPCs. Some important experiments and/or analysis are also lacking, such as CDX2 expression correlation with human myeloid and lymphoid malignancies, the regulation of Hox genes by Cdx2, and the detailed phenotypic and molecular analyses of lymphoid hematopoiesis. Therefore, although the current study provides some interesting findings and the experiments were carefully executed, the overall conclusions were not developed in sufficient depth in light of previous studies. There are several important remaining questions that need to be addressed, as detailed below.

Major points:

1. Since the Cdx2 KI overexpression by Scl-CreERT resulted in the development of both myeloid and lymphoid malignancies (Fig. 2), it will be important to analyze the effect of Cdx2 overexpression on normal lymphoid development (e.g. both T and B lineages and their progenitors).

Thank you for this question. As suggested, we have analysed the effect of Scl-CreERT:Cdx2 on lymphoid development. Scl-CreERT:Cdx2 mice exhibited reduced mature T cells and B cells in the peripheral blood (Figure 1B, and Supplementary Figure 2d). We now provide additional data regarding the early lymphoid development in these mice. The findings can be summarised as below.

Common lymphoid progenitors (CLP)

- Lineage- IL7R α + Kit^{low} Sca1^{low}: not significantly changed

T cell progenitors

- DN1 (TCR β + CD44+ CD25-): not significantly changed
- DN2 (TCR β + CD44+ CD25+): not significantly changed
- DN3 (TCR β + CD44- CD25+): not significantly changed
- DN4 (TCR β + CD44- CD25-): not significantly changed

B cell progenitors

- Pre-pro-B cells (B220+ CD19- cKit- IgM-): not significantly changed
- Pro-B cells (B220+ CD19+ cKit+ IgM-): significantly decreased in Scl:Cdx2
- Pre-B cells (B220+ CD19+ cKit- IgM-): significantly decreased in Scl:Cdx2
- Immature B cells (B220+ CD19+ cKit- IgM+): significantly decreased in Scl:Cdx2

We have added the following graphs (Supp Figure 1F-H) and additional text to the manuscript as below:

Main text:

In the BM, common lymphoid progenitors (CLP; lineage^{low}IL7Ra⁺) were unaffected by HSC expression of Cdx2 (Supplementary Figure 1F), as were double-negative T cell populations (DN1-4, Supplementary Figure 1G). However, a significant loss of B cell progenitors was observed (Supplementary Figure 1H), indicating a B lymphocyte differentiation block in Scl:Cdx2 BM.

Figure legend:

Frequency of (f) common lymphoid progenitors (CLP), (g) T cell progenitors (gated on TCRβ⁺; DN1, CD44+CD25⁻; DN2, CD44+CD25⁺; DN3, CD44-CD25⁺; DN4, CD44-CD25⁻), (h) B cells progenitors (Pre-pro-B, B220+ CD19⁻ cKit⁻ IgM⁻; Pro-B, B220+ CD19⁺ cKit⁺ IgM⁻; Pre-B, B220+ CD19⁺ cKit⁻ IgM⁻; immature B cells, B220+ CD19⁺ cKit⁻ IgM⁺) in WT or Scl:Cdx2 BM.

2. Detailed phenotypic analysis of the Cdx2 overexpression-induced lymphoid malignancies should be included. Similarly, detailed molecular analysis of gene expression and chromatin changes in the affected lymphoid populations should be included and compared to that in the HSPCs or myeloid cells. This line of investigation will reveal shared and unique target genes and associated cellular pathways responsible for Cdx2-mediated leukemia transformation in different hematopoietic lineages.

Thank you for this suggestion for additional analysis. We note that we have performed RNA sequencing on immunophenotypically defined stem cell populations (LSK⁺) however we were able to observe transcriptional priming that progressed during the development of specific leukemias.

We initially examined leukemias that had matched ATAC-seq and RNA-seq profiles and compared these to preleukemic samples and normal controls. Here, we were observed that RNAseq showed that preleukemic samples clustered closely with progenitor cell populations, however transformed samples gained transcriptional programs of the bulk leukemic cell population (Supplementary Figure 6E-F). Concordant with these data, ATACseq showed enrichment of early myeloid progenitor programs in pre-leukemia samples, with progressive acquisition of committed megakaryocyte erythroid progenitor chromatin architecture in erythroid leukemia, and lymphoid chromatin architecture in lymphoid leukemias, even though these cells retained a stem cell surface immunophenotype. Finally, we used RNAseq profiles of each Cdx2-expressing leukemia to identify differentially expressed genes that were upregulated in T-ALL (CS882) and B/T-ALL (CS252) but not other samples. Here, we found significant enrichment for genes deregulated upon transcription factor alteration in T lymphocytes and T-cell leukemia (p<0.05), again showing lymphoid priming within the stem cell populations. Tables resulting from this analysis are provided (Shared_Gene_DE.xlsx, enrichR.xlsx) and are now included in the manuscript as Supplementary Data.

3. Cdx2 overexpression significantly impaired LT-HSCs (Fig. 1H-I). Given the significant effects, it is surprising that the authors did not analyze other HSPC populations such as ST-HSCs and MPPs. It would be important to determine the molecular causes of LT-HSC exhaustion (e.g. impaired quiescence, proliferation, and/or differentiation).

Thank you for this important discussion point. After analysing differentiation within the LSK compartment (specifically STHSC and MPPs), we have determined that STHSC, like LTHSC, are also significantly depleted while MPP numbers are unchanged. We observed almost complete loss of LT-HSC in the *Cdx2* expressing mice, therefore we were unable to examine cell cycle within these cells. However, we do note that there is marked depletion of quiescent cells (Figure 5i) within *Cdx2*-expressing LSK. Therefore, we believe that the cellular mechanism of exhaustion is via enforced cell cycle entry.

The following text has been added to the results section.

“HSPC *Cdx2* expression led to depletion of long-term hematopoietic stem cells (LTHSCs [LKS+CD150+CD48-]) and short-term HSCs (STHSCs [LKS+CD150-CD48-]) in *Scl:Cdx2* BM (Figure 1H-I, Supplementary Figure 1M).”

“Multipotent progenitor (MPP [LKS+CD150-CD48+]) frequencies in *Scl:Cdx2* BM remained intact (data not shown), indicating a specific depletion of ST- and LTHSCs, the only cells with self-renewing potential²³, and implies that the cellular mechanism of LT-HSC exhaustion is via enforced cell cycle entry and loss of quiescence.”

4. *Cdx2* ChIP-seq should be performed and compared with ATAC-seq and RNA-seq datasets to define gene targets that may be directly regulated by *Cdx2*. The authors stated that ChIP-seq was not possible using rare HSPC populations. However, there are many modified ChIP-seq protocols that can work with thousands, hundreds of cells or even single cells, such as ChIPmentation and CUT&RUN (Schmidl et al., Nature Methods 12:963; Skene et al., Nature Protocols 13:1006; Hainer et al., Cell 177:1319).

Thank you for this suggestion. This issue was raised by all three Reviewers. We believe that this has now been addressed, please see the response to Reviewer 1 Q9 and Reviewer 2 Q3. The additional data is presented in the manuscript Figure 6B-E, Supplementary Figure 6 and presented as Reviewer Figure 4 (below R1 Q9) for convenience.

5. In Fig. 6E and Supplementary Fig. 5D, the authors included results that *Cdx2* overexpression increased chromatin accessibility at predicted *Cdx2* binding sites at *HoxA* and *HoxB* gene clusters, and *Cdx2* associates with the same binding sites in other cell types (ES and endoderm) by ChIP-seq. These results suggest that *Cdx2* may function to repress *HoxA* and *HoxB* gene expression in HSPCs in contrast to its activating roles in other cellular contexts. If validated these results could be highly significant that also explain the observed downregulation of *Hox* genes in *Cdx2* overexpressing HSPCs. Additional studies would be necessary to follow up on these interesting results.

Thank you for this comment. We agree that the repression of HoxA and HoxB cluster are interesting in this model. Both ATAC-seq and ChIP-Seq data demonstrate additional peaks in the HoxA and HoxB cluster and these peaks correspond with Cdx2 motifs (Figure 6E, reproduced below).

6. Given the observed phenotypes in both myeloid and lymphoid malignancies upon Cdx2 overexpression. It is important to also provide corroborative analyses of CDX2 expression in human myeloid and lymphoid malignancies. Is CDX2 overexpressed in human myeloid and lymphoid malignancies and correlated with poor survival and therapeutic responses? Is there difference in CDX2 expression with specific genetic lesions in myeloid and lymphoid malignancies? Does CDX2 expression positively or negatively correlate with HOX gene expression in different contexts?

While there is significant evidence for CDX2 overexpression in both myeloid and lymphoid malignancies, there are conflicting data that this is correlated with inferior survival or therapeutic response. Please also see the response to question 5 (above).

We suggest that CDX2 expression correlates differently with HOX expression in different contexts. For example, expression levels of CDX2 are comparable in acute lymphoid and acute myeloid leukemia samples (Thoene *et al.* 2009), however HOX deregulation is much less common in acute lymphoblastic leukemia (ALL) than AML. Furthermore, in embryogenesis, Cdx2 coordinates posterior development via Hox-independent mechanisms (Savory *et al.* 2009). These discussion points are now included in the manuscript.

Minor points:

1. The use of Flt3-ITD mouse model in combination of Cdx2 overexpression was not justified. Is CDX2 overexpression more commonly seen in human AML with FLT3-ITD mutations?

We agree that rationale for this strategy needs to be clarified. Reference to this has been added to the manuscript text as follows:

“Using the Beat AML trial cohort (Tyner *et al.* 2018), we found significant co-expression of CDX2 and FLT3 in AML patients, as well as increased CDX2 expression in FLT3-ITD-positive samples compared to FLT3-ITD-negative samples (data not shown). We therefore tested whether Scl:Cdx2 mice would accelerate development of AML when crossed with mice harboring Flt3-ITD, a common oncogene in AML”

2. Typo page 5 line 4, 'decreased' should be 'decreases'.

Thank you for bringing this to our attention. This has now been corrected.

REVIEWERS' COMMENTS:

Reviewer #1 (Remarks to the Author):

The authors have presented a much clearer and improved manuscript. In doing so they have answered all of my questions adequately and those of my fellow reviewers and, in doing so, are to be congratulated.

Reviewer #2 (Remarks to the Author):

This revised manuscript is significantly improved by the inclusion of CDX2 ChIP-seq data, as well as several other modifications to address reviewer concerns.

One issue that persists is the mechanism of secondary transformation. I appreciate their attempt to resolve this question, but I think that their response over-interprets the data, particularly with regard to Spi1 and Runx1 transcripts. There are no differences between control and Cdx2 HSPCs/AML for these genes based on Supplemental Fig 5F. This raises the question of whether differences shown in the heatmap (reviewer fig 6) are an artifact of the z-score normalization. My interpretation of Supp Fig 5F is that changes in Runx/PU.1 do not underlie secondary transformation, in contrast to what I'd postulated in my original review and the authors' current conclusion.

In any case, I'm willing to concede that this question may be more challenging than I'd initially anticipated, and it reasonably falls beyond the scope of the present manuscript. I do think that they should restate their paragraph and just focus on CEBP.

I have no other concerns.

Reviewer #3 (Remarks to the Author):

In the revised manuscript, the authors included new experiments and analyses to address this reviewer's original comments and the concerns of the other reviewers. The new phenotypic analyses of ST-HSCs, MPPs, lymphoid progenitors CLPs, T and B cell development in Cdx2 overexpression mouse model provide more refined information for the differential effect of Cdx2 expression on hematopoietic lineages. The Cdx2-FLAG-overexpression-based CHIP-seq results and further integration with RNA-seq and ATAC-seq provide additional support for the conclusions that CDX2 regulates gene transcription by acting at many distal elements. The authors also provided new evidences such as the analysis of CDX2 binding to the HoxA and HoxB clusters, and expression analysis of CDX2 in human myeloid and lymphoid leukemias, etc.

Overall the new data and textual revisions significantly enhanced the strength of the conclusions and addressed most of my original questions. This study provides an important mouse model and new mechanistic insights into the role of CDX2 in hematopoiesis and leukemia development. This reviewer has one remaining point for the revised manuscript that needs to be addressed, as following:

Given that the CDX2 CHIP-seq experiments were performed using retrovirally overexpressed CDX2-FLAG transgene in lineage-negative BM cells, there are several important caveats of these experiments that limited the strength of conclusions that one can draw. First, the lineage-negative cells are heterogeneous cells containing various stem and progenitor populations, thus the CHIP-seq signal represents an average of all populations which may be affected by changes in cell composition due to CDX2 overexpression. Second, overexpression of a transcription factor often leads to ectopic binding events not seen with endogenous proteins. As such, it is difficult to make strong conclusions based on these results and the subsequent integrative analyses with RNA-seq and ATAC-seq. It is appreciated that the CHIP-seq with endogenous CDX2 in low cell number purified populations might be technically challenging; therefore, the authors are advised to discuss the potential caveats of these experiments and tone down the relevant conclusions.

Reviewer comments:

Reviewer #1 (Remarks to the Author):

The authors have presented a much clearer and improved manuscript. In doing so they have answered all of my questions adequately and those of my fellow reviewers and, in doing so, are to be congratulated.

Thank you.

Reviewer #2 (Remarks to the Author):

This revised manuscript is significantly improved by the inclusion of CDX2 ChIP-seq data, as well as several other modifications to address reviewer concerns.

One issue that persists is the mechanism of secondary transformation. I appreciate their attempt to resolve this question, but I think that their response over-interprets the data, particularly with regard to Spi1 and Runx1 transcripts. There are no differences between control and Cdx2 HSPCs/AML for these genes based on Supplemental Fig 5F. This raises the question of whether differences shown in the heatmap (reviewer fig 6) are an artifact of the z-score normalization. My interpretation of Supp Fig 5F is that changes in Runx/PU.1 do not underlie secondary transformation, in contrast to what I'd postulated in my original review and the authors' current conclusion.

In any case, I'm willing to concede that this question may be more challenging than I'd initially anticipated, and it reasonably falls beyond the scope of the present manuscript. I do think that they should restate their paragraph and just focus on CEBP.

I have no other concerns.

Thank you for these remarks. We have re-drawn new graphs and modified the text such that only CEBP transcription factor genes are included for a more clarified analysis.

Reviewer #3 (Remarks to the Author):

In the revised manuscript, the authors included new experiments and analyses to address this reviewer's original comments and the concerns of the other reviewers. The new phenotypic analyses of ST-HSCs, MPPs, lymphoid progenitors CLPs, T and B cell development in Cdx2 overexpression mouse model provide more refined information for the differential effect of Cdx2 expression on hematopoietic lineages. The Cdx2-FLAG-overexpression-based ChIP-seq results and further integration with RNA-seq and ATAC-seq provide additional support for the conclusions that CDX2 regulates gene transcription by acting at many distal elements. The authors also provided new evidences such as the analysis of CDX2 binding to the HoxA and HoxB clusters, and expression analysis of CDX2 in human myeloid and lymphoid leukemias, etc.

Overall the new data and textual revisions significantly enhanced the strength of the conclusions and addressed most of my original questions. This study provides an important mouse model and new mechanistic insights into the role of CDX2 in hematopoiesis and leukemia development. This reviewer has one remaining point for the revised manuscript that needs to be addressed, as following:

Given that the CDX2 ChIP-seq experiments were performed using retrovirally overexpressed CDX2-FLAG transgene in lineage-negative BM cells, there are several important caveats of these experiments that limited the strength of conclusions that one can draw. First, the lineage-negative cells are heterogeneous cells containing various stem and progenitor populations, thus the ChIP-seq signal represents an average of all populations which may be affected by changes in cell composition due to CDX2 overexpression. Second, overexpression of a transcription factor often leads to ectopic binding events not seen with endogenous proteins. As such, it is difficult to make strong conclusions based on these results and the subsequent integrative analyses with RNA-seq and ATAC-seq. It is appreciated that the ChIP-seq with endogenous CDX2 in low cell number purified populations might be technically challenging; therefore, the authors are advised to discuss the potential caveats of these experiments and tone down the relevant conclusions.

Thank you for these comments. We have reworded our results regarding CDX2-ChIP on lineage-negative BM cells in order to accommodate the heterogeneous nature of this population.

“To overcome any dilution of binding signal of Cdx2-expressing HSPC, we sought to integrate Cdx2-FLAG ChIP-Seq data from lineage-negative cells with ATAC-Seq on LKS+ and publicly available CEBP α ChIP-Seq on GMP.”